# Does Continual Learning Meet Compositionality?
# New Benchmarks and An Evaluation Framework

**Weiduo Liao[1,2], Ying Wei[1*], Mingchen Jiang[1], Qingfu Zhang[1*], Hisao Ishibuchi[2*]**
[1]Department of Computer Science, City University of Hong Kong
[2]Department of Computer Science and Engineering, Southern University of Science and Technology
`weiduliao2-c@my.cityu.edu.hk`, {`yingwei, qingfu.zhang`}`@cityu.edu.hk`,
`jiangmingchen0129@gmail.com`, `hisao@sustech.edu.cn`

## Abstract

Compositionality facilitates the comprehension of novel objects using acquired concepts and the maintenance of a knowledge pool. This is particularly crucial for continual learners to prevent *catastrophic forgetting* and enable *compositionally forward transfer* of knowledge. However, the existing state-of-the-art benchmarks inadequately evaluate the capability of compositional generalization, leaving an intriguing question unanswered. To comprehensively assess this capability, we introduce two vision benchmarks, namely Compositional GQA (CGQA) and Compositional OBJects365 (COBJ), along with a novel evaluation framework called Compositional Few-Shot Testing (CFST). These benchmarks evaluate the *systematicity, productivity*, and *substitutivity* aspects of compositional generalization. Experimental results on five baselines and two modularity-based methods demonstrate that current continual learning techniques do exhibit somewhat favorable compositionality in their learned feature extractors. Nonetheless, further efforts are required in developing modularity-based approaches to enhance compositional generalization. We anticipate that our proposed benchmarks and evaluation protocol will foster research on continual learning and compositionality.

## 1 Introduction

Human understanding of the world relies on abstraction of concrete objects, allowing for the disentanglement of high-level understandings into low-level concepts. This ability, known as *compositionality*, is a crucial aspect of human intelligence as it addresses the *stability-plasticity dilemma* [11], and introduces a computationally efficient learning paradigm [3, 27]. By leveraging previously acquired concepts while acquiring new ones, individuals can effectively build and maintain knowledge when faced with a new task. For instance, accurate comprehension of varied bird wing shapes enables rapid identification of a new bird species by focusing on its wings. This compositional approach allows continual learners to achieve efficient knowledge reuse, benefiting not only in preventing *catastrophic forgetting* [37] of previous tasks (i.e., stability) but also in rapidly adapting to new tasks (i.e., plasticity).

There have been various approaches to continual learning (CL) that can be roughly categorized into three directions. Firstly, regularization-based methods [24, 33, 35] introduce regularization techniques to control the gradient update process for the current task. Secondly, replay-based methods [6, 4] store past tasks for later replay during training. Lastly, parameter-isolation-based methods, specifically modularity-based methods [28, 36, 43, 41], employ static or dynamic model architectures and allocate distinct parameters to each task. Nonetheless, it remains largely unclear

---

*Corresponding authors.

37th Conference on Neural Information Processing Systems (NeurIPS 2023) Track on Datasets and Benchmarks.

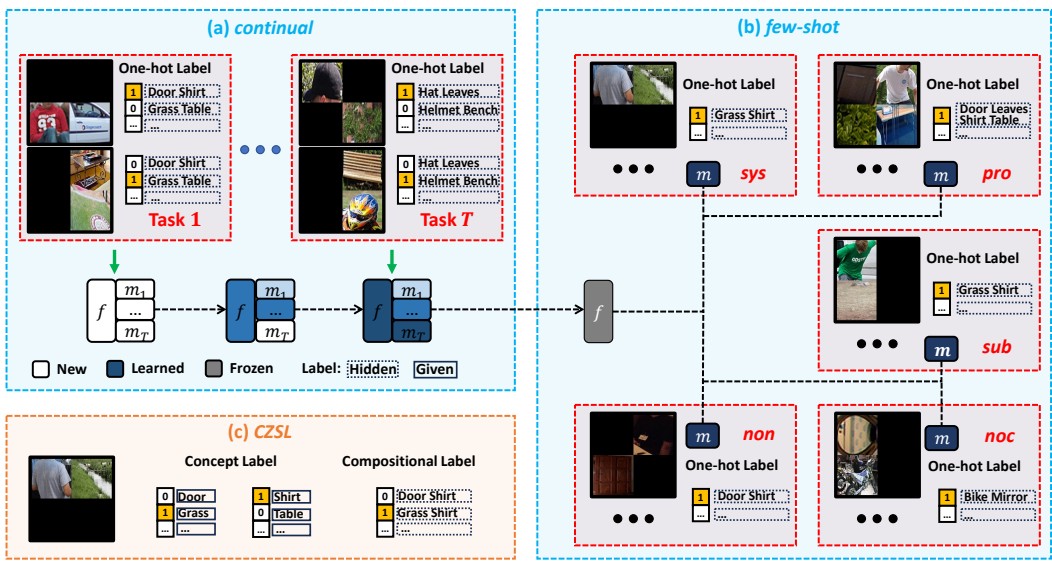

Figure 1: Process for CFST to test compositional generalization capability and comparison with CZSL. Taking CGQA as an example, after continually training $T$ tasks in (a), the feature extractor $f$ is frozen and we assign a new classifier specifically for each few-shot task. We use five types of few-shot tasks (b), which are introduced detailedly in section 4, to evaluate three aspects (i.e., *systematicity, productivity, substitutivity*) of compositionality, with the references of *non* and *noc*. CFST aims to challenge the compositional generalization ability of SOTA continual learners. Note that this protocol is suitable for class-IL and task-IL settings and we take the task-IL setting for illustration. Different from the CZSL setting in (c), our setting does not provide any concept-level supervision.

whether these methods truly satisfy compositionality. The current works have two main drawbacks. Firstly, the evaluation mainly focuses on catastrophic forgetting and overlooks forward transfer [35, 5]. The most relevant work to study compositionality is [41], which evaluates compositionality on a toy colored-MNIST dataset but only concentrates on color-shape composition. Secondly, the majority of existing CL benchmarks, such as Split-CIFAR10/100 and Split-MNIST, are insufficient to comprehensively evaluate compositionality. One potential alternative could be adapting benchmarks from the Compositional Zero-Shot Learning (CZSL) community for the CL setting. However, many of these benchmarks are either toy, synthetic, and lacking systematicity (e.g., CLEVR [21]), or restricted to simple object-attribute composition (e.g., UT-Zappos [56] and MIT-States [19]).

To address the aforementioned drawbacks and stimulate research on compositionality, we have developed two benchmarks: one synthetic yet systematic benchmark called CGQA, and one real-world benchmark called COBJ, both of which involve vision classification tasks. Additionally, we have designed an evaluation protocol called Compositional Few-Shot Testing (CFST), as illustrated in Figure 1), comprising a continual training phase where a learner accumulates knowledge from a sequence of tasks, followed by five few-shot testing phases. CFST aims to evaluate three compositional capabilities: 1) *Systematicity (sys)* — the ability to correctly predict outcomes for few-shot tasks with novel recombinations of familiar concepts; 2) *Productivity (pro)* — the ease of understanding combinations involving an increasing number of concepts; 3) *Substitutivity (sub)* — the speed at which the learner can acquire knowledge about slightly different concepts from those learned previously. Two additional evaluation aspects are considered: 1) *Non-novel (non)* — assessing the prediction accuracy for seen combinations of concepts; 2) *Non-compositional (noc)* — establishing a lower bound of accuracy for few-shot tasks by evaluating the learner with unseen concepts. CFST challenges the compositional generalization capability of state-of-the-art continual learners, with a primary focus on the feature extractor's ability to capture compositional concept features after continual training. Drawing inspiration from the field of natural language [18], we summarize the basic principles of compositional testing in vision. Firstly, the testing targets are novel recombinations of concepts found in the training set, ensuring that the continual learner has the opportunity to learn the meanings of these concepts (*solvability*). Secondly, while providing supervision for the testing task, it is crucial not to overwhelm the learner with excessive information, as they are expected to "quickly" understand the testing targets (*few-shot*). Finally, after acquiring sufficient knowledge during training, the feature

extractor is frozen to prevent further learning and overfitting during the testing phase (*frozen feature extractor*). Consequently, the continual learner must possess compositional generalization ability to bridge the systematic gap between training combinations and testing recombinations.

We assess the performance of five baseline methods and two parameter-isolation-based (modularity-based) methods on our benchmarks using CFST. The results indicate that the continual learning (CL) paradigm does introduce a decline in compositional performance on the concept level compared to the multi-task learning paradigm. Additionally, the efficiency of learning compositionality is highly influenced by the number of classes in each continual task. Furthermore, the state-of-the-art CL methods employing ResNet-18 backbone demonstrate some degree of compositional generalization capability in terms of Systematicity and Productivity, but struggle with Substitutivity. Conversely, methods utilizing the ViT backbone exhibit a preference for Substitutivity. Remarkably, we observe that the reusable knowledge acquired by modularity-based methods does not precisely correspond to the underlying concepts. It is clear that there is still a significant journey ahead for continual learners to develop interpretable compositionality.

## 2   Related Works

**Compositional benchmarks**   Compositionality is widely considered to be one of the most crucial elements in achieving a human-level understanding of the world. While compositionality has been extensively studied in the field of natural language, particularly in the context of local and global understanding [7], there is a need to evaluate the compositional capability of models. To address this, researchers have focused on creating datasets with specific split methods for training and testing purposes [7, 46, 18, 26, 23, 2, 22, 57, 10, 16]. However, the exploration of compositionality in the field of computer vision is still in its early stages. While some proposed benchmarks prioritize the use of toy or synthetic images [44, 41, 58], there is a lack of comprehensive evaluation on real-world datasets since most of them only test a limited aspect of compositionality (i.e., systematicity) [1, 31, 15, 14, 60, 34, 7, 20, 47].

**Continual learning**   Continual learning typically addresses the issue of *catastrophic forgetting* through various approaches, which can be broadly categorized into four families. **Regularization-based** methods [35, 33, 24] incorporate regularization techniques into the gradient updates to mitigate forgetting of previous tasks. **Replay-based** methods [6, 48, 59, 12] involve retraining on past task data or storing memory-efficient features to retain knowledge from previous tasks. **Parameter-isolation-based**, particularly hierarchical **modularity-based** methods [43, 30, 50, 9], focus on modularizing learners to enable compositional learning of tasks [38]. Building on the advancements of Vision Transformers [8], several **prompt-based** methods [53, 52, 51] aim to extract task-specific knowledge from a pre-trained backbone by using prompts. However, comparing these pre-trained methods to others may be somewhat unfair as they might have prior exposure to certain concepts or their combinations. Furthermore, only a limited number of studies (e.g., [41, 40, 50, 47]) explicitly evaluate their compositional capabilities, especially in the field of computer vision. For instance, [41] uses a toy dataset (color-MNIST) to evaluate simple color-shape compositions. [47] introduces a vision-and-language benchmark for continual concept learning tasks. In other works [32, 49, 1, 31, 15, 16, 25], including studies on compositionality outside the continual learning community, the datasets themselves possess compositional characteristics, but their evaluation may not rigorously test novel combinations.

## 3   Preliminaries

**Vision continual classification tasks**   Continual learning focuses on a sequence of $T$ vision classification tasks, each involving an $N$-way classification. For each task $t \in \mathcal{T} = [1 \dots T]$, there is a corresponding training set $D_t = \{(x_t, y_t)\}$, where $x_t \in \mathcal{X}$ represents the input image and $y_t \in \mathcal{Y}$ is the corresponding class label. Here, $N$ denotes the number of classes within each individual task. A continual learner is a neural network comprised of two main components. Firstly, there is a feature extractor $f(x) : \mathcal{X} \to \mathcal{L}$, which maps input images to an extracted feature space. Secondly, a multi-task classifier $m(f(x), t) : \mathcal{L} \times \mathcal{T} \to \mathcal{Y}$ is present. This classifier takes the extracted features from the feature extractor, $f(x)$, as well as the task identifier, $t$, as inputs and produces a prediction for the corresponding task. Here, $\mathcal{Y} \subset \mathbb{R}$ represents the label space, which will be discussed further in

the subsequent paragraph. It is important to note that tasks are sequentially presented to the continual learner, thereby deviating from the assumption of independent and identically distributed (i.i.d.) data.

*Remark* 3.1. This setting corresponds to the task-incremental (task-IL) scenario. Additionally, we also investigate the class-incremental (class-IL) setting, where the task identifier $t$ is not provided. Consequently, the multi-task classifier $m(f(x), t)$ simplifies to $m(f(x))$.

**Concept-based categories** From a human learning perspective, it is essential for representations to be compositional, meaning that a global understanding of an object is formed by combining all its semantic sub-parts [39]. For instance, people learn to recognize others by memorizing characteristics such as their hair color and mouth size. To assess whether the model is capable of learning compositional representations in images, we initially define the components, referred to as *concepts*, that we want the model to learn explicitly. Specifically, we create a pool of candidate concepts, denoted as $\mathcal{C} = \{c^i\}_{i=1}^M$, where $M$ represents the number of concepts. The vector $c^{1:M} = (c^1, \ldots, c^M) \in \{0,1\}^M$ indicates the presence of these concepts, where $\{0,1\}^M$ denotes the $M$-fold Cartesian product $\{0,1\} \times \cdots \times \{0,1\}$, and $c^i = 1$ indicates the presence of the $i$-th concept, while 0 denotes its absence. Further details regarding the continual training stage and evaluation stages will be discussed in section 4.

*Remark* 3.2. Although we define a multi-hot concept space for data generation, it is important to note that $y$ remains a single image-level label, and no concept label is provided to the candidate learner. This distinction is illustrated in Figure 1, where categories such as {Door, Shirt} and {Grass, Shirt} are considered as two completely separate categories despite sharing the concept *Shirt*. As a result, the learner must deduce the underlying compositional concepts within an image. This represents the key difference between our label space definition and compositional zero-shot learning (CZSL), as depicted in Figure 1c.

## 4 Compositional Few-Shot Testing Protocol: CFST

We will now elucidate the precise methodology for assessing the compositionality of a learner through the utilization of the Compositional Few-Shot Testing (CFST) protocol, as illustrated in Figure 1.

**Concept factorization** We adopt *attribute factorization* [54, 29] to model the data generation process by using a latent variable $z$ since it accurately captures the different concept distributions between training and testing tasks. Denoting the joint distribution of the image $x$ and the corresponding hidden concepts as $p(c^{1:M}, x)$, we can write

$$z \sim p(z), \quad c^i \sim p(c^i|z), \quad x \sim p(x|z), \quad p(c^{1:M}, x) = p(c^{1:M}) \int p(x|z)p(z|c^{1:M})\mathrm{d}z.$$

We split the label space $\mathcal{Y}_{tr}, \mathcal{Y}_{nv} \subset \mathcal{Y}, \mathcal{Y}_{tr} \bigcap \mathcal{Y}_{nv} = \emptyset$ using different marginal distributions, $p(c^{1:M}) \neq p_{tr}(c_{tr}^{1:M}) \neq p_{nv}(c_{nv}^{1:M})$, and keeping the same conditional generative model $p(x|c^{1:M}) = \int p(x|z)p(z|c^{1:M})\mathrm{d}z$ for different phases, thus, artificially introducing compositional distribution shifts. Note that, $c_{tr}^{1:M}, c_{nv}^{1:M}$ are the corresponding concepts in $\mathcal{Y}_{tr}, \mathcal{Y}_{nv}$, respectively. Without loss of generality, we bound the maximal number of activated concepts in $\mathcal{Y}_{tr}$, such that $p_1$-norm $\|c_{tr}^{1:M}\|_1 \leq M_{tr}$. The principles guiding the evaluation are summarized as follows:

1. **Few-shot tasks**: After continually training with $T$ $N$-way tasks with $p_{tr}(c_{tr}^{1:M})$, we expect this learner should have acquired sufficient knowledge in the concept pool $\mathcal{C}$ and be able to recognize novel concept combinations $y_{nv} \in \mathcal{Y}_{nv}$ with $p_{nv}(c_{nv}^{1:M})$ using only a small number of support samples. Thus, we adopt *few-shot* tasks, in which each concept combination $y_{nv}$ has a small support set $D_s$. Proper recognition of underlying compositional concepts is crucial to avoid severe overfitting.

2. **Solvability**: All $M$ concepts in $\mathcal{C}$ should be encountered at least once in the continual training tasks $\{D_t\}_{t=1}^T$. Additionally, the learner should have sufficient exposure to learn the concepts in various combinations. We take the label {Grass, Shirt} with concept *Grass* in Figure 1 as an example, *Grass* should be seen in at least one training task, e.g., {Grass, Table}. This assumption aligns with the general idea that standard gradient-based models cannot generalize to nonlinear functions without observing enough diverse examples [55].

3. **Novelty**: Few-shot tasks should only contain novel recombination of concepts not seen in the training set, i.e., $\mathcal{Y}_{nv} \bigcap \mathcal{Y}_{tr} = \emptyset$. The presence of any seen label would negatively impact the

evaluation of the learner's compositional ability. In the above example, $\{D_t\}_{t=1}^T$ should not contain any {Grass, Shirt} sample.

4. **Frozen feature extractor**: We recommend not further updating the feature extractor $f$ using the support samples $D_s$ in the few-shot evaluation task. Only the classifier $m$ is updated to assess whether $f$ extracts the expected features related to the underlying concepts in the few-shot task. Empirically, a non-frozen feature extractor attempting to learn from a few-shot task leads to a noticeable reduction in performance, as shown in Appendix.

Based on the mentioned principles, we present five different few-shot testing schemes for the continual learner:

**Systematicity Novel Testing (*sys*)**   To examine the comprehension of learners on novel combinations of concepts in $\mathcal{C}$, we design $T_{nv}$ $N$-way $K$-shot tasks. Each task includes $N$ concept combinations $y_{sys}$ randomly selected from $\mathcal{Y}_{sys}$, with $K$ samples for each $y_{sys}$. Here, $\mathcal{Y}_{sys} \subset \mathcal{Y}_{nv}$ and the corresponding $\|c_{sys}^{1:M}\|_1 \leq M_{tr}$. It should be noted that $\mathcal{Y}_{nv}$ does not include any previously seen combinations in $\mathcal{Y}_{tr}$, ensuring the fulfillment of the *novelty* requirement.

**Productivity Novel Testing (*pro*)**   The acquired knowledge should be able to generalize effectively to complex images that contain a greater number of previously encountered concepts. After learning about {Door, Shirt}, {Grass, Table}, and {Hat, Leaves}, the learner should have no trouble recognizing {Door, Leaves, Shirt, Table}. To assess this ability, we construct few-shot tasks that involve more concepts per task compared to the *sys* testing. Specifically, we consider another label space $\mathcal{Y}_{pro} \subset \mathcal{Y}_{nv}$ where the candidate label $y_{pro} \in \mathcal{Y}_{pro}$ satisfies the condition $\|c_{pro}^{1:M}\|_1 > M_{tr}$.

**Substitutivity Novel Testing (*sub*)**   Concepts can possess specific attributes, such as the color of a *Shirt* being either red or green. Given knowledge acquired from images of red shirts (or shirts that are not green) and an understanding of *Grass* being green, learners are expected to quickly recognize that a green shirt is still a *Shirt*, as it combines the attribute of green color with the object of a shirt. In order to assess compositional understanding at the attribute level, we maintain the same distribution of concepts $p_{sub}(c_{sub}^{1:M}) = p_{sys}(c_{sys}^{1:M})$, but utilize different conditional generative models $p_{sub}(x|c_{sub}^{1:M}) \neq p(x|c^{1:M})$. It is important to note that this attribute-based testing should be observed in relation to other concepts during continual training, allowing learners the opportunity to recognize it (*solvability*). This distinction is the primary difference between our *sub* approach and the domain-incremental setting.

The aforementioned three novel few-shot schemes provide a comprehensive evaluation of compositionality. In addition, we will introduce two reference schemes for the purpose of comparison and analysis.

**Non-novel Testing (*non*)**   To assess the few-shot testing performance of learners on trained combinations, we create few-shot tasks where the label is taken from $\mathcal{Y}_{tr}$ rather than $\mathcal{Y}_{sys}$. This particular setting intentionally violates the *novelty* assumption. The average test accuracy on these *non* tasks serves as a complement to the average test accuracy on the tasks involving continually trained combinations. Additionally, comparing the results with other few-shot novel methods (such as *sys*) allows us to determine if the learner possesses the specific compositionality.

**Non-compositional Testing (*noc*)**   We introduce *noc* tasks to evaluate learners' few-shot testing performance on combinations that involve unseen concepts. In this case, the label $y_{noc}$ is selected from $\mathcal{Y}_{noc}$, and the condition $\|c_{noc}^{1:M}\|_1 \leq M_{tr}$ applies, with $\mathcal{C}_{noc} \bigcap \mathcal{C} = \emptyset$. These tasks follow the normal continual learning scheme with $T + 1$ tasks, but the last task is a few-shot task and the feature extractor remains frozen. This particular setting intentionally violates the *solvability* assumption. The average test accuracy on *noc* tasks serves as a useful metric to indicate the lower limit of few-shot testing performance.

## 5   Proposed Benchmarks

The existing continual learning (CL) community lacks a comprehensive benchmark specifically designed to evaluate compositionality. In order to address this, we can adapt existing benchmarks for Compositional Zero-shot Learning (CZSL) for this purpose. However, directly transforming CZSL

benchmarks into CL versions, as done with Split-CIFAR100, is not appropriate. This is because some CZSL benchmarks are considered to be toy benchmarks, such as CLEVR [21]. Furthermore, most CZSL benchmarks do not emphasize systematic compositionality, as they primarily focus on simple object-attribute compositions, such as UT-Zappos [56] and MIT-States [19]. Simply combining multiple benchmarks is also not suitable, as the concepts involved in different benchmarks might vary, resulting in a lack of cross-benchmark concept combination.

Our objective in developing benchmarks is to incorporate a wide range of concept combinations alongside a substantial number of instances for each concept. Thus, we first introduce a synthetic benchmark known as Compositional GQA (CGQA), derived from GQA [17] (License CC BY 4.0). CGQA offers comprehensive combinations of concepts and a substantial number of concept instances. Then, to evaluate continual learners using real-world images, we also propose a real-world benchmark called Compositional Objects365 (COBJ), sourced from Objects365 [45] (License CC BY 4.0). Due to the limitations of space, please refer to the Appendix for image examples and additional benchmark statistics.

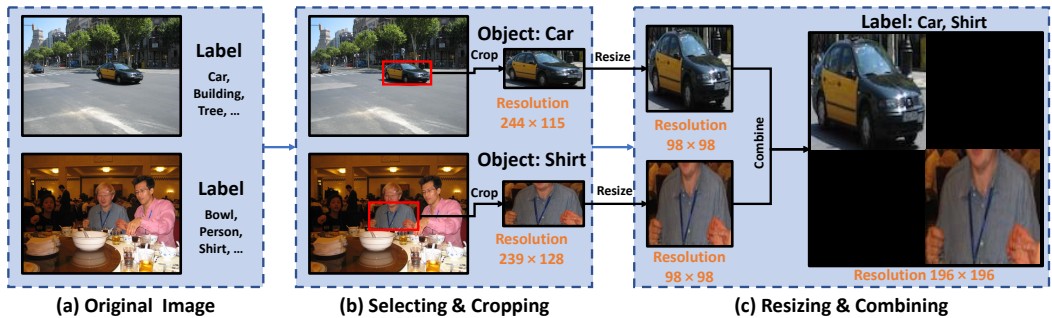

Figure 2: Example of CGQA construction.

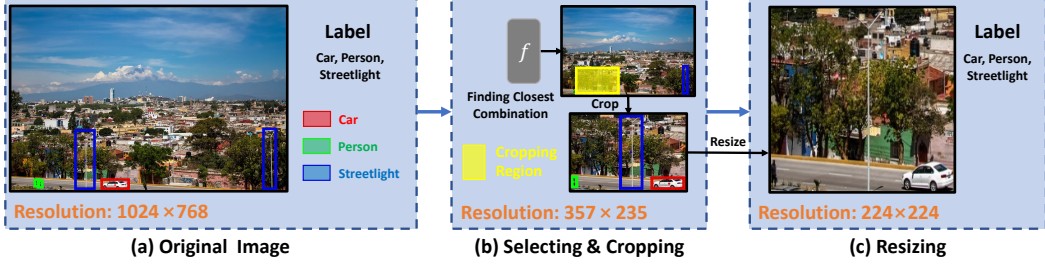

Figure 3: Example of COBJ construction.

**Construction of CGQA**   Since the available number of annotations in GQA is inadequate to construct the compositional benchmark using the original images, we have chosen to artificially combine suitable objects and create new images for our purposes. The process is illustrated in Figure 2. Initially, we crop the image using the bounding box of the selected object and then resize the cropped region to a resolution of $98 \times 98$ pixels. Next, we randomly position the processed object image within a $2 \times 2$ grid. Any unoccupied grid spaces are filled with black pixels to ensure a complete composition.

**Construction of COBJ**   The images in Object365 are of high resolution and contain multiple instances of classes, annotated with bounding boxes (as shown in Figure 3a). To conveniently utilize these images, we need to resize them. However, resizing high-resolution images directly can lead to a loss of object resolution, which may have a detrimental effect on the model's performance. Therefore, we employ a strategy where we select a specific bounding box within each object set to crop and generate a new image. The process is explained in Figure 3. Subsequently, we perform the resizing operation, allowing the object to occupy a larger proportion within the image and providing more pixels for analysis. The selection of bounding boxes follows the principle of minimizing the total distance between their respective center points. This process ensures that the closest bounding boxes are chosen. By using the cropped region shown in yellow in Figure 3b, we reduce the proportion of irrelevant parts and maximize the preservation of the original object's information.

Table 1: Results on CGQA. Accuracy $(\%)\pm 95\%$ confidence intervals $(\%)$ over 300 tasks are reported for few-shot testing phases. Task-IL settings denote with a postfix "*".

| Methods | $A_{con}$ | $A_{sys}$ | $A_{pro}$ | $A_{sub}$ | $H_n$ | $A_{non}$ | $A_{noc}$ | $H_r$ | $H_a$ |
|---|---|---|---|---|---|---|---|---|---|
| MultiTask | 83.61 | $88.14 \pm 0.62$ | $85.94 \pm 0.65$ | $69.67 \pm 0.75$ | 80.35 | $91.55 \pm 0.51$ | $40.04 \pm 0.99$ | 55.71 | 68.28 |
| Finetune | 8.38 | $64.73 \pm 0.78$ | $65.43 \pm 0.73$ | $61.26 \pm 0.67$ | 63.75 | $68.54 \pm 0.80$ | $40.32 \pm 0.72$ | 50.77 | 57.84 |
| ER | 19.78 | $\mathbf{71.38 \pm 0.75}$ | $70.11 \pm 0.64$ | $64.32 \pm 0.69$ | 68.46 | $\mathbf{77.27 \pm 0.67}$ | $40.98 \pm 0.72$ | 53.56 | 61.60 |
| GEM | 8.56 | $66.56 \pm 0.77$ | $\mathbf{73.47 \pm 0.61}$ | $62.80 \pm 0.71$ | 67.33 | $72.65 \pm 0.73$ | $41.64 \pm 0.72$ | 52.94 | 60.72 |
| LwF | 9.11 | $71.22 \pm 0.74$ | $73.28 \pm 0.61$ | $\mathbf{68.74 \pm 0.66}$ | $\mathbf{71.03}$ | $76.56 \pm 0.66$ | $\mathbf{48.69 \pm 0.77}$ | $\mathbf{59.52}$ | $\mathbf{65.93}$ |
| EWC | 8.22 | $64.99 \pm 0.78$ | $\mathbf{73.47 \pm 0.64}$ | $63.25 \pm 0.66$ | 66.95 | $69.03 \pm 0.74$ | $41.38 \pm 0.71$ | 51.75 | 59.91 |
| RPSnet | $\mathbf{33.45}$ | $59.80 \pm 0.83$ | $60.26 \pm 0.72$ | $59.75 \pm 0.74$ | 59.94 | $64.22 \pm 0.81$ | $45.09 \pm 0.64$ | 52.98 | 56.95 |
| MultiTask* | 92.17 | $82.16 \pm 0.68$ | $84.22 \pm 0.60$ | $71.82 \pm 0.75$ | 79.01 | $86.44 \pm 0.69$ | $44.07 \pm 0.95$ | 58.38 | 69.23 |
| Finetune* | 72.46 | $70.32 \pm 0.73$ | $72.62 \pm 0.63$ | $66.33 \pm 0.69$ | 69.66 | $75.32 \pm 0.70$ | $43.26 \pm 0.73$ | 54.95 | 62.92 |
| ER* | $\mathbf{76.05}$ | $\mathbf{71.37 \pm 0.70}$ | $72.67 \pm 0.69$ | $\mathbf{66.80 \pm 0.63}$ | $\mathbf{70.19}$ | $\mathbf{76.28 \pm 0.66}$ | $45.61 \pm 0.77$ | $\mathbf{57.09}$ | $\mathbf{64.29}$ |
| GEM* | 21.60 | $69.44 \pm 0.75$ | $73.14 \pm 0.67$ | $66.61 \pm 0.66$ | 69.63 | $73.31 \pm 0.71$ | $\mathbf{45.97 \pm 0.74}$ | 56.51 | 63.71 |
| LwF* | 73.19 | $69.61 \pm 0.77$ | $\mathbf{74.22 \pm 0.63}$ | $65.00 \pm 0.65$ | 69.40 | $75.25 \pm 0.69$ | $42.52 \pm 0.72$ | 54.34 | 62.48 |
| EWC* | 71.10 | $69.40 \pm 0.79$ | $72.37 \pm 0.65$ | $65.10 \pm 0.67$ | 68.83 | $74.99 \pm 0.72$ | $42.82 \pm 0.77$ | 54.51 | 62.29 |
| MNTDP* | 68.98 | $47.27 \pm 0.91$ | $47.41 \pm 0.86$ | $47.50 \pm 0.85$ | 47.39 | $52.28 \pm 0.81$ | $32.29 \pm 0.83$ | 39.93 | 44.09 |

# 6 Experimental Studies

In this section, we conducted experiments using SOTA continual learning algorithms on our proposed compositional benchmarks, namely CGQA and COBJ. The purpose of these comprehensive computational experiments is to address the following research questions: (1) Can our proposed benchmarks accurately evaluate the learners' ability to extract compositional features from a sequence of tasks? (2) Can five different few-shot testing schemes provide insightful perspectives on the learners' compositionality? (3) Do modularity-based approaches demonstrate superior few-shot testing accuracy, indicating their efficacy in extracting compositional features?

## 6.1 Experiment Settings

**Compared Algorithms**  We evaluate five baseline algorithms, including: Finetune, which trains a continual learner sequentially for all tasks; regularization-based methods: GEM [35], LwF [33], and EWC [24]; a replay-based method: ER [6]. Additionally, we also evaluate two SOTA hierarchical compositional modularity approaches: MNTDP [50] (MNTDP-D), which dynamically incorporates new modules or reuses existing frozen modules to improve performance on the current task; RP-Snet [43], which randomly selects modules in each layer and employs regularization and experience replay for assistance. When evaluating these methods in the class-incremental learning (class-IL) setting, where the task identifier is unknown, we use a single-head classifier. In the task-incremental learning (task-IL) setting, where the task identifier is known, we employ a multi-head classifier (denoted by a postfix "*"). All the algorithms mentioned above are evaluated using a ResNet-18 backbone [13]. For detailed hyper-parameter settings and additional results using a ViT backbone [8], please refer to the Appendix.

**Evaluation Metric**  During the continual training phase, we report the **Average test accuracy** $A_{con}$ on all seen tasks after completing $T = 10$ tasks. For the five different few-shot testing phases, namely *sys*, *pro*, *sub*, *non*, and *noc*, we report the **Average test accuracy** for each test (i.e., $A_{sys}, A_{pro}, A_{sub}, A_{non}$, and $A_{noc}$) on $T_{nv} = 300$ tasks. It is worth noting that the $A_{sub}$ metric is not applicable for COBJ due to the absence of attribute information in the source benchmark Objects365. We denote the prediction accuracy of the multi-task classifier $m$ as $\Delta(m(f(x), t), y)$. The corresponding testing dataset for task $t$ is denoted as $D_t^{te}$. The Average test accuracy is calculated as $\frac{1}{T_{nv}} \sum_{t=1}^{T_{nv}} \mathbb{E}_{(x,y) \sim D_t^{te}}[\Delta(m(f(x), t), y)]$. Additionally, we employ the **Harmonic mean** $H_n = 3/(1/A_{sys} + 1/A_{pro} + 1/A_{sub})$ among *sys*, *pro*, and *sub*, $H_r = 2/(1/A_{non} + 1/A_{noc})$ between *non* and *noc*, and $H_a = 5/(1/A_{sys} + 1/A_{pro} + 1/A_{sub} + 1/A_{non} + 1/A_{noc})$ to show the compositionality, reference capability, and overall few-shot testing capability, respectively. The Harmonic mean places emphasis on achieving balanced performance across all few-shot cases simultaneously, measuring the model's ability to perform well across different scenarios.

Table 2: Results on COBJ. Accuracy $(\%) \pm 95\%$ confidence intervals $(\%)$ over 300 tasks are reported for few-shot testing phases. Task-IL settings denote with a postfix "*".

| Methods | $A_{con}$ | $A_{sys}$ | $A_{pro}$ | $H_n$ | $A_{non}$ | $A_{noc}$ | $H_r$ | $H_a$ |
|---|---|---|---|---|---|---|---|---|
| MultiTask | 39.73 | $72.70 \pm 1.55$ | $67.11 \pm 1.86$ | 69.79 | $83.38 \pm 1.36$ | $57.52 \pm 1.81$ | 68.08 | 68.93 |
| Finetune | 6.37 | $43.06 \pm 1.28$ | $41.89 \pm 1.29$ | 42.46 | $46.96 \pm 1.51$ | $46.12 \pm 1.36$ | 46.54 | 44.41 |
| ER | 7.67 | $46.51 \pm 1.48$ | $45.22 \pm 1.38$ | 45.86 | $57.58 \pm 1.85$ | $42.81 \pm 1.32$ | 49.11 | 47.43 |
| GEM | 5.57 | $46.11 \pm 1.41$ | $43.99 \pm 1.33$ | 45.03 | $49.51 \pm 1.57$ | $46.07 \pm 1.47$ | 47.73 | 46.34 |
| LwF | 6.73 | $55.10 \pm 1.54$ | $50.12 \pm 1.54$ | 52.49 | $59.37 \pm 1.61$ | $51.40 \pm 1.55$ | 55.10 | 53.76 |
| EWC | 6.53 | $42.73 \pm 1.19$ | $41.00 \pm 1.32$ | 41.85 | $46.69 \pm 1.48$ | $41.21 \pm 1.34$ | 43.78 | 42.79 |
| RPSnet | **21.93** | $\textbf{73.26} \pm \textbf{1.24}$ | $\textbf{67.89} \pm \textbf{1.44}$ | **70.47** | $\textbf{78.82} \pm \textbf{1.27}$ | $\textbf{68.22} \pm \textbf{1.24}$ | **73.14** | **71.78** |
| MultiTask* | 75.73 | $55.72 \pm 1.75$ | $49.84 \pm 1.81$ | 52.62 | $59.21 \pm 1.89$ | $53.43 \pm 1.74$ | 56.17 | 54.34 |
| Finetune* | 44.37 | $45.67 \pm 1.39$ | $44.18 \pm 1.38$ | 44.91 | $47.92 \pm 1.54$ | $47.80 \pm 1.57$ | 47.86 | 46.34 |
| ER* | 64.80 | $\textbf{54.40} \pm \textbf{1.73}$ | $\textbf{51.33} \pm \textbf{1.69}$ | **52.82** | $58.91 \pm 1.73$ | $48.83 \pm 1.54$ | 53.40 | 53.11 |
| GEM* | 36.53 | $45.57 \pm 1.39$ | $45.50 \pm 1.39$ | 45.53 | $51.93 \pm 1.64$ | $46.40 \pm 1.36$ | 49.01 | 47.21 |
| LwF* | 61.03 | $53.77 \pm 1.63$ | $50.06 \pm 1.76$ | 51.84 | $\textbf{60.54} \pm \textbf{1.58}$ | $\textbf{54.53} \pm \textbf{1.61}$ | **57.38** | **54.47** |
| EWC* | 53.87 | $45.63 \pm 1.46$ | $46.12 \pm 1.47$ | 45.88 | $51.16 \pm 1.59$ | $48.86 \pm 1.48$ | 49.98 | 47.84 |
| MNTDP* | **71.80** | $51.89 \pm 1.41$ | $49.16 \pm 1.56$ | 50.49 | $57.20 \pm 1.59$ | $48.52 \pm 1.36$ | 52.50 | 51.48 |

## 6.2 Results

We report the performance results of our CGQA benchmark in Table 1 and the results of the COBJ benchmark in Table 2. For a detailed analysis of the COBJ benchmark results under different settings of $T$, please refer to the Appendix.

**Multi-task baselines outperform CL methods in both continual tasks and compositional generalization performance** In both class-IL and task-IL settings, MultiTask(*) demonstrates the highest performance on $A_{con}$ and $H_n$, which is as expected since this method benefits from i.i.d. training task sequences. To visualize the learned concept region, we employ the CAM [61]. The results in the Appendix for both CGQA and COBJ demonstrate that the proposed $H_n$ accurately evaluates learners' performance in extracting compositional features. Notably, CL methods generally do not outperform MultiTask(*) on $H_n$, revealing that forgetting also contributes to the degradation of compositionality. This observation highlights the urgent question of how to equip learners with compositionality for the continual learning community.

**Continual tasks prefer modularity-based methods** Among the baselines, RPSnet achieves a significantly larger margin on $A_{con}$, particularly on COBJ, indicating that the reused knowledge in the modules can effectively mitigate forgetting. While MNTDP performs poorly on CGQA, it outperforms the other methods on the real-world benchmark COBJ. This difference can be attributed to the fact that forgetting is not as significant on CGQA as it is on COBJ, especially in the task-IL setting. Therefore, simply freezing modules does not provide efficient reusable (compositional) knowledge for the novel recombination of seen concepts. The second-best method on $A_{con}$ is ER(*), particularly in the class-IL setting, demonstrating the effectiveness of the memory buffer.

**Modularity-based methods show no superiority on compositionality** In terms of $H_n$, another winning method alongside ER* is LwF, although its performance on $A_{con}$ is not outstanding. This suggests that the feature extractor learned by LwF is excellent at extracting compositional features, but the classifier fails to make correct predictions. The phenomenon of *prediction bias* is further discussed in the Appendix, with CAM visualization. Regularization-based mechanisms like GEM, LwF, and EWC struggle to solve this classifier issue. As a result, only replay-based methods (such as ER and RPSnet) show improved performance. Unfortunately, RPSnet and MNTDP* do not demonstrate superiority on CGQA, indicating that the reused knowledge is not sufficiently compositional for handling novel combinations of seen concepts. Freezing the modules for old tasks may not be appropriate for new tasks unless the modules have enough compositionality. Further efforts are required to enhance modularity-based methods to bring benefits to compositionality. Interestingly, on COBJ, these two methods show competitive performance compared to others, with RPSnet performing the best. This indicates their superiority in dealing with real-world tasks.

**Five few-shot tests exhibit a consistent order:** $A_{non} > A_{pro} > A_{sys} > A_{sub} > A_{noc}$   Generally, the performance on *non* is the best, which is expected because models tend to favor the seen combinations and may struggle with compositional gaps to some extent. The performance gap between *sys* and *pro* is acceptable, but it is larger for *sub*. This is because *sub* requires the composition of attributes of concepts to form unseen recombination, demanding a higher-level compositional ability compared to *sys* and *pro*. Interestingly, when models have a ViT backbone (in the Appendix), we observe contrary results: $A_{sub} > A_{pro} > A_{sys}$. It can be attributed to the fact that the selected attributes for concepts in *sub* primarily pertain to texture attributes (e.g., color). This observation aligns with the analysis in [42] that the multi-head attention layer in ViT has a bias towards shape attributes, while the convolution layer has a bias towards texture attributes. Therefore, models with a ResNet-18 backbone are more sensitive to the texture difference between *continual* and *sub*. Regarding accuracy on *pro*, it is generally slightly better than on *sys*, except for MultiTask and ER, indicating that having more concepts can somewhat aid in understanding the images. The overall lower accuracy on the single-head baselines (class-IL setting) compared to their multi-head counterparts (task-IL setting), except for LwF, demonstrates that the single-head classifier makes it difficult for the feature extractor to retain compositional knowledge.

**Concept-level forgetting**   We conducted a case study to investigate the phenomenon of forgetting with regard to concepts. Following a continual training phase, we evaluated the learned models on four few-shot task sets: *non(f)*, which consists of **seen** combinations of **freshly** learned concepts in the last task; *non(o)*, which consists of **seen** combinations of **old** concepts from previous tasks; *sys(f)*, which consists of **novel** combinations of **freshly** learned concepts in the last task; and *sys(o)*, which consists of **novel** combinations of **old** concepts from previous tasks. If the model experiences forgetting on the concept level, we would expect a small gap between *non(f)* and *sys(f)* and a large gap between *non(o)* and *sys(o)*. The construction of these two task sets is detailed in the Appendix. We also report the related scores $S(o) = (A_{sys(o)} - A_{non(o)})/A_{non(o)}$ and $S(f) = (A_{sys(f)} - A_{non(f)})/A_{non(f)}$ to clearly show the performance gaps. The average test accuracy and the related score are shown in Figure 4.

Our observations reveal that $A_{non(f)} > A_{non(o)}$ and $A_{sys(f)} > A_{sys(o)}$, which is as expected since fresh concepts are newly updated in the last continual task. Additionally, $S(f) > S(o)$, indicating that the learner has better systematicity compositional generalization capacity for fresh concepts than for old concepts. These observations demonstrate that the learner indeed learns concepts in a composition-wise manner and forgetting occurs primarily with old concepts. On the other hand, Multi-Task, which is regarded as the upper bound, experiences the smallest performance drop on both fresh and old tasks, confirming our expectation that the learner jointly learns all continual tasks. This observation also

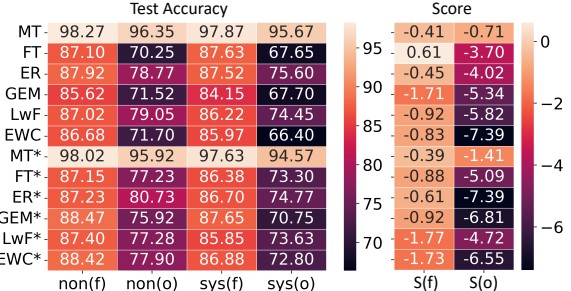

Figure 4: Fresh tasks *vs* old tasks on CGQA. MT stands for MultiTask and FT stands for Finetune. **Take away**: fresh concepts suffer little performance drop when testing on novel combinations (sys(f)) while old concepts (sys(o)) on the contrary.

suggests that the continual learner experiences forgetting not only on old tasks but also on old concepts. To visualize the learned concepts, we used CAM; please refer to the Appendix for more details.

**Sample efficiency for learning compositionality**   To investigate the number of training samples required for a continual learner to acquire the ability of compositional generalization, we vary the number of training samples for each class in the CGQA continual training tasks. It is intuitive that a small number of training samples not only hinders novel combinations (i.e., *sys, pro, sub*), but also affects the seen combinations (i.e., *non*). Examples of *sys* and *non* are depicted in the left and middle figures of Figure 5, respectively. The complete results can be found in the Appendix. Moveover, we utilize the related score, denoted as $S(sys)$. It is calculated as $(A_{sys} - A_{non})/A_{non}$, where a larger positive $S(sys)$ indicates better compositional generalization ability on Systematicity, while a smaller negative value indicates poor handling of novel combinations of seen concepts by the learner. The results are presented in the right figure of Figure 1. For almost all methods, $S(sys)$ reaches

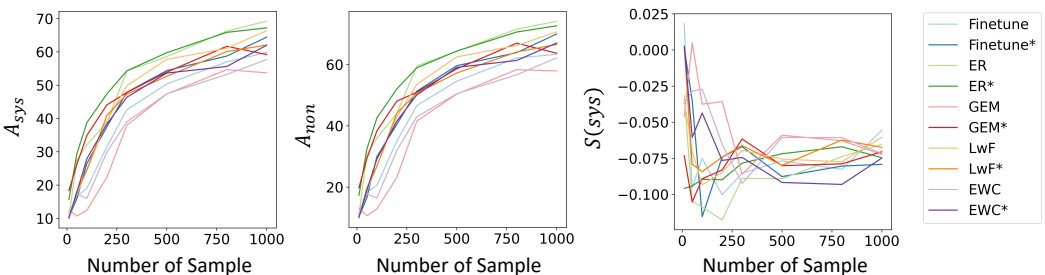

Figure 5: Varying number of training samples on CGQA. **Take away**: Accuracy is improved with more training samples for both *sys* and *non*. Related scores become smooth and stable if more than 300 training samples for each class. This fact shows that 300 samples are needed for learning compositionality.

convergence when trained with more than 300 samples for each class. Beyond this threshold, the ratio of improvement for *sys* to *non* remains unchanged. Consequently, we can conclude that learners need 300 samples for each class to learn compositionality in CGQA.

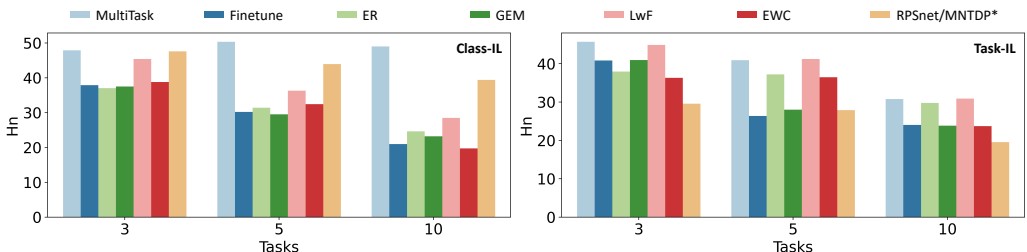

Figure 6: Number of continual tasks *vs* Hn on 10-way COBJ testing tasks. **Take away**: A larger number of continual tasks (smaller ways for each task) hinders compositionality.

**Varying number of continual training tasks** As the number of training tasks increases, the number of classes in each task decreases while the total number of classes remains constant. Consequently, compared to a single task that includes all the classes, the learner requires a smaller number of compositional features to distinguish classes in each individual task. However, these features may be insufficient for future compositional tasks. To evaluate the learners, we train them on three (10-way), five (6-way), and ten (3-way) COBJ tasks respectively, and then assess their performance on 300 10-way few-shot *sys* and *pro* tasks. We visualize the results using $H_n$, as shown in Figure 6. The observation aligns with the aforementioned understanding. For more detailed information, please refer to the Appendix.

## 7  Conclusion

In this paper, we present two benchmarks, CGQA and COBJ, which offer ample samples for CL. Additionally, we introduce CFST, a novel evaluation protocol that enables comprehensive assessment of compositionality in three key aspects: *systematicity, productivity*, and *substitutivity*. The construction of our benchmarks is based on publicly available sources, ensuring the absence of sensitive data and ethical suitability for research purposes. Experimental results demonstrate that our protocol provides valuable insights for SOTA CL methods. However, modularity-based approaches do not exhibit superiority when subjected to compositional testing, highlighting the need for significant advancements in this area. We believe that our benchmarks and evaluation protocol will inspire researchers to prioritize compositionality during the development of continual learners.

**Limitation and future work** While our work specifically focuses on local compositionality, future studies will explore the global scopes of compositionality, extensively examined in other fields such as natural language processing. Although our candidate compositions of concepts are limited, they sufficiently serve the purpose of testing compositionality in continual learners. Moving forward, the combination of multiple benchmarks will be considered to form a more comprehensive evaluation framework.

## Acknowledgments and Disclosure of Funding

This work was supported by National Natural Science Foundation of China (Grant No. 62250710163, 62250710682), Guangdong Provincial Key Laboratory (Grant No. 2020B121201001), the Program for Guangdong Introducing Innovative and Enterpreneurial Teams (Grant No. 2017ZT07X386), The Stable Support Plan Program of Shenzhen Natural Science Fund (Grant No. 20200925174447003), Shenzhen Science and Technology Program (Grant No. KQTD2016112514355531).

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
