# Does Continual Learning Meet Compositionality? New Benchmarks and An Evaluation Framework

**Weiduo Liao[1,2], Ying Wei[1*], Mingchen Jiang[1], Qingfu Zhang[1*], Hisao Ishibuchi[2*]**
[1]Department of Computer Science, City University of Hong Kong
[2]Department of Computer Science and Engineering, Southern University of Science and Technology
weiduliao2-c@my.cityu.edu.hk, {yingwei, qingfu.zhang}@cityu.edu.hk,
jiangmingchen0129@gmail.com, hisao@sustech.edu.cn

## Contents

37th Conference on Neural Information Processing Systems (NeurIPS 2023) Track on Datasets and Benchmarks.

# A   Discussion

**How catastrophic forgetting happens**    The common assumption of an independent and identically distributed (i.i.d.) training data distribution usually does not hold in continual learning. In the extreme case, classes seen in the previous tasks may never appear afterward. Inevitably, a model trained with a naive strategy (i.e., sequentially passing all tasks only once) will encounter *catastrophic forgetting* [22]; that is, it drastically forgets the learned knowledge for old tasks upon learning the new task. The reasons are roughly two-fold: (1) the feature extractor updates itself so that it can not extract some important features for old tasks; (2) the single-head classifier for the class-incremental setting will bias the prediction to the classes in the current task (a.k.a. prediction bias). We give evidence empirically by using our evaluation of compositionality on the feature extractor and also CAM visualization (in section E.3).

**Another example of compositional learning**    Let's consider a set of fine-grained dog species recognition tasks. A learner can learn general dog concepts (e.g., body, head, and feet shapes) and class-specific concepts (e.g., the small size of the Chihuahua) when recognizing the Chihuahua. After that, it can reuse these general concepts for other species, while only needing to learn new class-specific concepts (e.g., the fur color of the Alaskan) further, rather than building up a specific understanding totally from scratch. The above example clearly states the importance of compositionality when acquiring knowledge from the continually changing world. However, the question of whether continual learning really meets compositionality is still under exploration.

**Relationship between our protocol with compositional zero-shot learning**    Our setting is the most general and difficult one that only the compositional label is provided. Thus learners need to learn the hidden concepts in one image. Compositional zero-shot learning (CZSL) provides both concept and compositional labels to help recognize novel combinations of concepts (i.e., zero-shot prediction). Here we generalize CZSL to the case that *objects* and *states* are all as *concepts* (thus the combination can be any number of concepts from any concept set), while the standard CZSL [32, 2, 28] targets learning *object* and *state* labels separately (thus the combination of concepts only happens between two different concept sets).

On the other hand, our few-shot testing tasks are somehow related to the evaluation in CZSL. From the task-level perspective, the non-novel test (i.e., *non*) is to evaluate the seen (trained) classes, and three novel tests (i.e., *sys, pro, sub*) are to evaluate the unseen (untrained) classes in a close-world setting. If we jointly consider all few-shot tests, it is in an open-world setting.

**Relationship between our few-shot testing with few-shot learning**    Although we adopt few-shot tasks here, the whole process is not related to few-shot learning. In the few-shot testing phase, only the classifier is learned with the support set, so that the testing accuracy can directly reflect how well the feature extractor understands the hidden concepts in the task.

**Relationship between compositionality with forward transfer**    In [21], forward transfer is defined as the average test accuracy improvement on the next task between testing on the current model and testing on a random initialization. However, random initialization is not an appropriate choice

for comparison. In this paper, we provide compositional few-shot tasks to explicitly evaluate the performance of the feature extractor on recognizing previously learned concepts and eliminate the effect of the classifier. The result directly reflects how well the current learner can adapt to the new compositional tasks. Thus, we exactly test the compositional forward transfer.

**Relationship between compositionality with forgetting**   An ideal compositional feature extractor reuses concept-level knowledge to extract interpretable image representations. They can be directly used for the current task, thus, suffer less gradient update. In this ideal case, the forgetting is eliminated. From our experiments, we indeed observe the concept-level forgetting by case studies and find that forgetting tends to decrease when the learner has a better compositional generalization capability.

**Relationship between substitutivity test with domain-IL setting**   In the substitutivity test, the few-shot tasks contain novel re-combinations of seen concepts and the concepts have a domain shift. However, it is not the domain-IL setting, since the labels (i.e., novel re-combinations of seen concepts) are unseen before, and the feature extractor is frozen in the substitutivity test. We adopt this setting to investigate whether the learner can transfer the attribute-level knowledge onto other concepts.

**Difficulties for models to learn compositionality**   Firstly, we would like to highlight that increasing the number of classes in one task tends to improve the compositionality in the CL setting, which is empirically demonstrated in the experiments of varying numbers of classes. Here we further give another intuitive example. let us consider one class $A$ in which all training samples are on a white background and one class $B$ in which all training samples are on a black background instead. The model can easily distinguish $A$ and $B$ according to the background (i.e., the so-called geometric skews [26]), but not according to the features of the class instance itself. However, if there is another class $C$, which is also in black background. In this tri-classification task (distinguishing $A$, $B$, and $C$), an ideal model should focus on the feature of the instance itself but not the background. This is one of the difficulties: **distribution bias** on samples, that some beneficial features (e.g., background) may be good for the classification, but not good for understanding the class (in a compositional way). Another difficulty is **entanglement** of the labels. We provide the labels in a relative way that the label of $A$ is '0' and of $B$ is '1', but not their true textual meanings (e.g., *white paper* and *green leaves*). The concept information is entangled and embedded into the label, thus, it is hard for the model to tell which visual features capture the corresponding concepts (i.e., *white* refers to the color feature and *paper* refers to the texture feature). We hope our understanding of this issue can inspire researchers to focus more on compositionality and design excellent continual learners.

# B   Additional Related Works

## B.1   Compositional Zero-shot Learning

In the field of compositional zero-shot learning, researchers fully explore the learning of object-attribute composition with the provided object and attribute labels. [25] proposes a set of regularizers for object-attribute composition and treats attributes as operations. [32] independently learns objects and attributes with prototypes. [2] treats object and attribute labels as causes of an image, rather than its effects. [13] learns object-state dependency by self-attentions. However, we aim to evaluate whether continual learners understand tasks in a compositional way. Thus, these methods do not fit our setting in that the continual learners have to learn hidden concepts with only combined labels and should not be pre-trained. [42] unites EBM modules for concept learning and does not require concept labels, but only focus on toy vision datasets.

## B.2   Continual Learning

Typically, continual learning aims to handle *catastrophic forgetting* mainly in the three families. **Regularization-based** methods include GEM [21], that projects the current gradient direction not to hurt the past tasks, LwF [19], that uses knowledge distillation loss to prevent the learner from large distribution shifts of old classifiers' responses on the current task, and EWC [14], that constrains important parameters related to the old tasks to alleviate forgetting. **Replay-based** methods include ER [3], that rehearses some past tasks' samples from a replay buffer when learning from the current

task. Further, [35, 43, 8] replays hidden features rather than raw samples for memory-efficient learning.

Parameter-isolation-based (mainly hierarchical **modularity-based**) methods focus on modularizing learners to learn tasks in a compositional way [24]. Since compositionality is a crucial capability to address *catastrophic forgetting*, these studies have been active in recent years in a continual learning manner. The state-of-the-art approaches differ according to how they compose modules for the specific task. For **single-layer** composition, [23] and ELLA [33] propose to linearly combine modules and Model Zoo [31] ensembles modules using AdaBoost. Since the expressive ability of the single-layer composition is limited and concepts are intuitively to be hierarchical, the following methods consider **multi-layer** (hierarchical) composition. First, the *parallel ordering* structure maintains a set of candidate modules for each layer. RPSnet [30], [16], HOUDINI [37], and MNTDP [38] choose one module in each layer to form a chaining structure. On the other hand, PathNet [7] selects a certain number of modules from the candidates in each layer discretely and sum (or concatenate) the outputs to pass to the next layer, while AdaComp [44] and LMC[27] learn soft-weights for candidate modules in one layer as the overall output. Secondly, the *permuted ordering* structure assembles a set of modules in a task-specific order (e.g., [25]). Another *soft ordering* structure allows all modules to take parts in all layers in a soft-weighting way (e.g., [23]).

Interestingly, although the above methods are proposed to learn connections among tasks and maximize the functional reuse of modules, only a small number of them (e.g., [27, 25, 38]) explicitly test their compositional abilities with toy datasets, especially in the vision domain. In [18, 36, 1, 17, 10, 11, 15] including compositional works outside the continual learning community, the datasets themselves are compositional but the evaluation is not strictly testing the novel combination.

Profiting from the development of Vision Transformer [6], several **prompt-based** methods [41, 40, 39] aim to extract task-specific knowledge from a pretrained backbone by prompting. However, comparing with such pretrained methods are relatively unfair since they may have seen some concepts or concept combinations before.

## C  Benchmark Details

In this section, we present all the details of our proposed benchmarks, including some image examples, the construction process, and training/testing class/concept orders. Our benchmarks and code are available at `https://github.com/CityU-LANTERN/CFST`.

### C.1  Compositional GQA (CGQA)

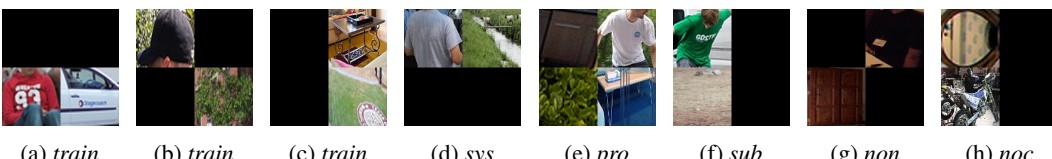

|   (a) *train*   |   (b) *train*   |   (c) *train*   |   (d) *sys*   |   (e) *pro*   |   (f) *sub*   |   (g) *non*   |   (h) *noc*   |

Figure 1: Image examples for CGQA. The labels for *continual*, i.e., *train* (a-c) are (Door, Shirt), (Hat, Leaves), and (Grass, Table), respectively. For *sys* (d), the testing combination is (Grass, Shirt), in which grass and shirt are all seen in *continual*. For *pro* (e), the testing combination is (Door, Leaves, Shirt, Table). For *sub* (f), (Grass, Shirt) only contains brown grass and green shirt. For *non* (g), the trained combination (Door, Shirt) is tested. For *noc* (h), (Bike, Mirror) contains unseen bike and mirror concepts.

The source dataset is GQA [12] (License CC BY 4.0). GQA is a real-world visual question-answering dataset, which provides a comprehensive object set with useful meta-information, e.g., attributes.

### C.1.1  Construction of CGQA

Our constructed CGQA benchmark consists of a continual training phase and five few-shot testing phases, i.e., *sys*, *pro*, *sub*, *non*, and *noc*. We first pick $M = 21$ objects with as many attributes as possible and larger than 30 pixels in width and height utilizing the provided bounding boxes. These

objects form our concept pool $\mathcal{C}$ in the continual training phase. For each object, we select one attribute specifically for *sub*. For example, color can be a good candidate since it is commonly present on different objects which easily guarantees the solvability requirement. Then, we put those object instances which have the corresponding selected attributes into a *attr* set. After that, we split the remaining object instances into *train*, *val*, *test*, and *fewshot* sets and make sure the learner will not see exactly the same object instances in different sets. Among these sets, *train*, *val*, and *test* are used in the continual training phase, and *fewshot* is used in the few-shot testing phases.

### C.1.2 Continual Training Phase

We construct 100 random and different object combinations as the $\mathcal{Y}_{tr}$ and each $y_i \in \mathcal{Y}_{tr}$ consists of $M_{tr} = 2$ objects in $\mathcal{C}$ with randomly assigned positions in the image (e.g., left-bottom and right-bottom), as shown in Figure 1a-1c. In these combinations, all objects in $\mathcal{C}$ should be witnessed as balanced as possible. We generate 1000 (*train*), 50 (*val*), 100 (*test*) image instances for each combination from the corresponding *train*, *val*, *test* sets, respectively. Generally, we construct $T = 10$ tasks, each with $N = 10$ object combinations $\{y_i\}_{i=1}^{N}$.

### C.1.3 Few-shot Testing Phase

In this phase, we have five different testing schemes, i.e., *sys*, *pro*, *sub*, *non*, and *noc*. We will introduce all these schemes in detail as follows:

1. **Systematicity Novel Testing (*sys*)** We construct another set $\mathcal{Y}_{sys}$ containing 100 novel object combinations different from $\mathcal{Y}_{tr}$. The example image is shown in Figure 1d. Different from the large *train* set, we only need to generate 100 image instances for each combination from the *fewshot* set. For clarity, we also name this image set as *sys*.
2. **Productivity Novel Testing (*pro*)** In the *pro* test, more concepts are in one image than that in the continual training phase. Specifically as shown in Figure 1e, 3 or 4 objects (concepts) are sampled with randomly assigned positions in one image. We sample 100 image instances for each combination from the *fewshot* set and name this image set as *pro*.
3. **Substitutivity Novel Testing (*sub*)** For the *sub* test, object instances are sampled from *attr* set, which contains only the objects with selective attributes, as shown in Figure 1f. We sample 100 image instances for each combination and name this image set as *sub*.
4. **Non-novel Testing (*non*)** Different from the continual training phase in which training objects are from the *train* set, in the *non* test, we sample object instances from the *fewshot* set to construct 100 image instances and name this image set as *non*. The example image is shown in Figure 1g.
5. **Non-compositional Testing (*noc*)** We construct another novel concept pool $\mathcal{C}_{noc}$, with other 200 objects in GQA rather than the seen objects in $\mathcal{C}_{tr}$. Further, we design 100 object combinations $\mathcal{Y}_{noc}$ and each with 100 image instances and name this image set as *noc*. The example images are shown in Figure 1h.

From the obtained image sets (i.e., *sys*, *pro*, *sub*, *non*, and *noc*), we construct $T_{nv}$ $N$-way $K$-shot tasks for each set by randomly selecting images, and $T_{nv} = 300, N = 10, K = 10$. The reported average test accuracy is used as the metric to analyze the compositional generalization capability of the learner after continual training.

### C.1.4 Training Order

Unless stated otherwise, the 21 training concepts for CGQA are: *plate, shirt, building, sign, grass, car, table, chair, jacket, shoe, flower, pants, helmet, bench, pole, leaves, wall, door, fence, hat, shorts.* We choose these 21 concepts because they have numerous instances in the original dataset. The combinations of concepts need to be sufficiently explored among the 10 tasks. The continual training order can be generated randomly. For the sake of analysis, we fix the continual training order in the experiments. The following shows the order:

- Task 1: *(fence, flower), (door, grass), (leaves, shirt), (grass, table), (shoe, shorts), (hat, table), (leaves, wall), (chair, grass), (door, shoe), (fence, helmet)*;
- Task 2: *(chair, sign), (grass, shorts), (hat, plate), (pole, shirt), (grass, pants), (pants, shoe), (pole, wall), (bench, chair), (helmet, plate), (leaves, shoe)*;
- Task 3: *(bench, shorts), (flower, pole), (chair, helmet), (pants, shorts), (helmet, shorts), (helmet, shoe), (hat, jacket), (hat, shorts), (jacket, shoe), (fence, wall)*;

- Task 4: *(bench, helmet), (hat, shirt), (bench, sign), (plate, wall), (grass, plate), (helmet, pole), (door, leaves), (bench, pants), (grass, jacket), (jacket, pole)*;
- Task 5: *(car, jacket), (building, plate), (helmet, leaves), (pants, shirt), (car, leaves), (bench, leaves), (fence, pants), (bench, shirt), (fence, grass), (building, jacket)*;
- Task 6: *(fence, plate), (car, helmet), (car, shorts), (grass, leaves), (jacket, shirt), (chair, shirt), (plate, sign), (bench, jacket), (leaves, sign), (chair, shoe)*;
- Task 7: *(flower, shirt), (building, chair), (plate, shorts), (building, leaves), (chair, hat), (fence, pole), (grass, sign), (building, grass), (hat, shoe), (bench, wall)*;
- Task 8: *(car, flower), (bench, door), (bench, hat), (bench, building), (bench, table), (hat, sign), (shirt, wall), (door, fence), (door, plate), (pole, table)*;
- Task 9: *(flower, pants), (shoe, sign), (helmet, shirt), (leaves, plate), (hat, wall), (grass, shoe), (plate, shirt), (pants, wall), (fence, leaves), (chair, pole)*;
- Task 10: *(car, sign), (car, pants), (flower, helmet), (building, hat), (car, shirt), (helmet, sign), (flower, wall), (door, pole), (leaves, shorts), (fence, shorts)*.

### C.1.5 Concept Statistics

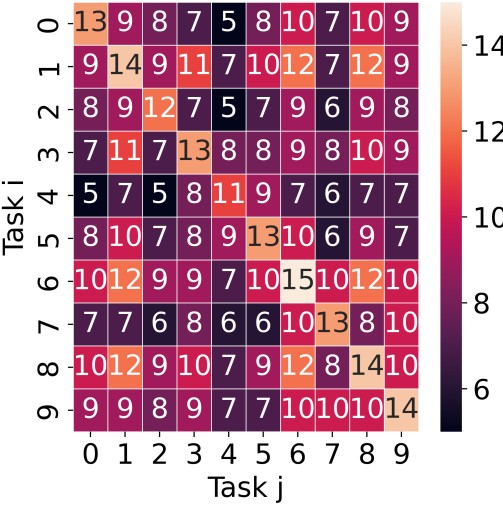

Figure 2: Number of the same concepts for the continual training tasks in CGQA.

The concepts in the continual training tasks are shown as follows:

- Task 1: *chair, door, fence, flower, grass, hat, helmet, leaves, shirt, shoe, shorts, table, wall*;
- Task 2: *bench, chair, grass, hat, helmet, leaves, pants, plate, pole, shirt, shoe, shorts, sign, wall*;
- Task 3: *bench, chair, fence, flower, hat, helmet, jacket, pants, pole, shoe, shorts, wall*;
- Task 4: *bench, door, grass, hat, helmet, jacket, leaves, pants, plate, pole, shirt, sign, wall*;
- Task 5: *bench, building, car, fence, grass, helmet, jacket, leaves, pants, plate, shirt*;
- Task 6: *bench, car, chair, fence, grass, helmet, jacket, leaves, plate, shirt, shoe, shorts, sign*;
- Task 7: *bench, building, chair, fence, flower, grass, hat, leaves, plate, pole, shirt, shoe, shorts, sign, wall*;
- Task 8: *bench, building, car, door, fence, flower, hat, plate, pole, shirt, sign, table, wall*;
- Task 9: *chair, fence, flower, grass, hat, helmet, leaves, pants, plate, pole, shirt, shoe, sign, wall*;
- Task 10: *building, car, door, fence, flower, hat, helmet, leaves, pants, pole, shirt, shorts, sign, wall*.

We visualize numbers of the same concepts pair-wisely in Figure 2. CGQA provides comprehensive combinations of training concepts, which guarantees that the learner has the chance to learn compositionality.

### C.1.6 Few-shot Testing Order

Then, we list the concept combinations for few-shot tests:

- **sys**: The 100 *systematicity* test combinations are *(bench, car), (bench, fence), (bench, grass), (bench, plate), (bench, pole), (bench, shoe), (building, car), (building, door), (building, fence), (building, helmet), (building, pants), (building, shirt), (building, shorts), (building, table), (building, wall), (car, chair), (car, door), (car, fence), (car, grass), (car, hat), (car, plate), (car, pole), (car, shoe), (car, table), (car, wall), (chair, door), (chair, fence), (chair, flower), (chair, jacket), (chair, leaves), (chair, pants), (chair, plate), (chair, shorts), (chair, table), (chair, wall), (door, flower), (door, hat), (door, helmet), (door, jacket), (door, pants), (door, shirt), (door, shorts), (door, sign), (door, table), (door, wall), (fence, hat), (fence, jacket), (fence, shirt), (fence, shoe), (fence, sign), (fence, table), (flower, grass), (flower, hat), (flower, jacket), (flower, leaves), (flower, shoe), (flower, shorts), (flower, sign), (flower, table), (grass, hat), (grass, helmet), (grass, pole), (grass, shirt), (grass, wall), (hat, helmet), (hat, leaves), (hat, pants), (helmet, jacket), (helmet, pants), (helmet, table), (jacket, leaves), (jacket, pants), (jacket, plate), (jacket, shorts), (jacket, sign), (jacket, table), (jacket, wall), (leaves, pants), (leaves, pole), (pants, plate), (pants, pole), (pants, sign), (pants, table), (plate, pole), (plate, shoe), (plate, table), (pole, shoe), (pole, shorts), (pole, sign), (shirt, shoe), (shirt, shorts), (shirt, sign), (shirt, table), (shoe, table), (shoe, wall), (shorts, table), (shorts, wall), (sign, table), (sign, wall), (table, wall).*
- **pro**: The 100 *productivity* test combinations are *(bench, building, door, sign), (bench, car, fence), (bench, car, flower), (bench, car, flower, leaves), (bench, car, helmet), (bench, car, jacket, pole), (bench, chair, grass, shirt), (bench, chair, pole), (bench, door, fence, hat), (bench, door, hat), (bench, door, leaves, pole), (bench, door, plate, pole), (bench, door, shoe, sign), (bench, fence, table, wall), (bench, flower, hat, jacket), (bench, grass, pants), (bench, grass, pants, pole), (bench, hat, jacket, shorts), (bench, helmet, pants, wall), (bench, helmet, table, wall), (bench, leaves, shirt), (bench, leaves, sign), (bench, pants, shoe), (bench, pole, sign), (bench, shorts, sign, wall), (building, car, fence, shorts), (building, car, hat, plate), (building, car, leaves), (building, chair, fence, shorts), (building, chair, helmet), (building, door, pants, shorts), (building, flower, helmet, shoe), (building, grass, hat, shorts), (building, grass, pants, sign), (building, helmet, pants, pole), (building, leaves, table), (building, pants, shoe), (building, plate, sign), (building, shirt, shorts, table), (car, door, fence, helmet), (car, door, hat, pole), (car, door, shorts, table), (car, fence, helmet), (car, fence, pants), (car, flower, jacket, table), (car, flower, shirt), (car, grass, hat, shirt), (car, grass, helmet), (car, helmet, shoe, shorts), (car, jacket, plate, pole), (car, jacket, shorts), (car, leaves, shorts), (car, shoe, shorts), (chair, flower, shoe), (chair, hat, helmet, plate), (chair, hat, jacket), (chair, helmet, shoe, sign), (chair, jacket, plate), (chair, jacket, shoe), (chair, leaves, pole, shorts), (chair, pole, sign), (chair, shorts, table), (chair, sign, wall), (door, flower, leaves, shorts), (door, hat, helmet, shoe), (door, hat, shirt, table), (door, helmet, jacket, shirt), (door, helmet, table, wall), (door, jacket, shirt, shoe), (door, jacket, table), (door, jacket, wall), (door, plate, shirt), (fence, grass, helmet, wall), (fence, helmet, leaves), (fence, helmet, pants, pole), (fence, helmet, shoe, sign), (fence, jacket, pants), (flower, grass, pants, pole), (flower, grass, pole, table), (flower, hat, jacket, shoe), (flower, hat, leaves, pole), (flower, helmet, pants, pole), (flower, helmet, plate), (flower, jacket, shorts), (flower, pants, shorts), (flower, shoe, shorts), (hat, helmet, jacket), (hat, helmet, shoe), (hat, jacket, pole), (hat, leaves, sign), (hat, leaves, table, wall), (helmet, jacket, pants), (jacket, shirt, shoe, wall), (leaves, table, wall), (pants, pole, shirt), (pants, pole, shoe), (pants, sign, wall), (pants, table, wall), (plate, shoe, sign), (pole, shorts, wall).*
- **sub**: The *substitutivity* tests consider the selective attributes for concepts: *grass*: brown, *shirt*: green, *plate*: blue, *pants*: white, *leaves*: brown, *fence*: black, *helmet*: white, *shoe*: black, *jacket*: blue, *car*: red, *table*: white, *chair*: black, *flower*: yellow, *pole*: wood, *building*: brown, *hat*: blue, *bench*: metal, *wall*: brick, *door*: glass, *sign*: round, *shorts*: white.
- **non**: The test combinations for the non-novel test are the same as those in the continual train phase.
- **noc**: The unseen concepts are *jeans, eyes, lady, orange, snow, cow, keyboard, wire, motorcycle, couch, frisbee, beak, tower, kite, bridge, window, hands, beach, guy, horse, wing, banana, word, player, spoon, rock, bed, sauce, screen, woman, street, airplane, bowl, sock, eye, cloud, tree, numbers, label, train, floor, animal, dirt, ball, uniform, bike, cat, gravel, logo, paper, laptop, desk, suit, zebra, bag, van, glasses, letter, food, platform, pot, sweater, child, pavement, sticker, man, hand, foot, shelf, face, counter, ear, road, cabinet, mane, wheel, fork, can, onion, fur, leg, elephant, goggles, roof, bush, blanket, curtain, sky, boy, umbrella, boot, mouth, broccoli, tail, bus, hill, trash, post, cup, collar, trunk, truck, frame, book, dog, ground, ceiling, windshield, cheese, cake, arrow, head, bear, ocean, vehicle, carrot, ring, tie, finger, fruit, refrigerator, sun, sidewalk, napkin, street light, paw, lid, girl, coat, seat, water, plant, sheep, glove, sneakers, cone, toilet, tire, tag, spots, faucet, stick, lamp, racket, phone, mirror, donut, number, wrist, nose, sand, knife, neck, box, branch, dress, meat, bird, hair, field, basket, vase, picture, people, boat, cap, house, pizza, sink,*

*clock, room, arm, watch, bottle, vest, pillow, container, camera, paint, mountain, giraffe, bread, apple, horn, drawer, cord, bicycle, computer, feet, towel, backpack, stone, flag, wetsuit, balcony, carpet, candle, snowboard, scarf, necklace.* From these unseen concepts, we randomly generate 100 non-compositional test combinations: *(floor, vehicle), (pizza, book), (girl, ball), (pizza, coat), (apple, floor), (meat, player), (boy, sand), (hand, hair), (ring, faucet), (zebra, carrot), (kite, mane), (vest, bottle), (bowl, cake), (sheep, tree), (fur, frisbee), (nose, carpet), (onion, fork), (pot, sink), (neck, watch), (roof, towel), (bridge, dog), (bicycle, foot), (sink, curtain), (label, racket), (frame, windshield), (knife, ear), (broccoli, van), (bear, racket), (toilet, lady), (ground, collar), (zebra, flag), (bed, sheep), (wing, food), (pavement, ground), (hand, wetsuit), (stone, beak), (wrist, guy), (phone, frame), (room, phone), (ring, ground), (sock, number), (sheep, motorcycle), (face, cheese), (hill, vase), (elephant, laptop), (wrist, can), (shelf, scarf), (broccoli, boot), (eyes, clock), (mouth, vase), (kite, eye), (wheel, stick), (wheel, wrist), (ocean, watch), (ball, mane), (box, broccoli), (airplane, word), (wire, room), (label, refrigerator), (window, wetsuit), (fork, banana), (camera, hand), (cord, lady), (letter, zebra), (ocean, tire), (street light, necklace), (branch, plant), (tie, truck), (horse, girl), (pillow, motorcycle), (street light, foot), (train, backpack), (napkin, collar), (candle, bowl), (bush, pillow), (glasses, cow), (sneakers, collar), (dirt, clock), (desk, bowl), (airplane, pavement), (towel, food), (boat, wire), (hill, dress), (apple, guy), (sweater, banana), (logo, laptop), (scarf, camera), (elephant, onion), (post, phone), (trash, wheel), (motorcycle, mountain), (phone, lady), (cup, arrow), (ocean, broccoli), (bridge, cheese), (snow, ring), (elephant, bike), (child, sky), (bush, napkin), (room, dog).*

### C.1.7    Alternative: Compositional PartImageNet (CPIN) Benchmark

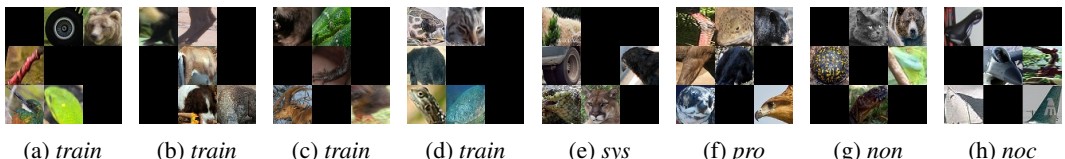

| (a) *train* | (b) *train* | (c) *train* | (d) *train* | (e) *sys* | (f) *pro* | (g) *non* | (h) *noc* |

Figure 3: Image examples for CPIN. The labels for *continual*, i.e., *train* (a - d) are (Bird Head, Snake Head, Car Tier, Bird Foot, Ursidae Head), (Bovidae Body, Primates Hand, Lacertilia Body, Canidae Head, Bovidae Foot), (Mustelidae Foot, Canidae Foot, Lacertilia Foot, Snake Head, Bovidae Head), and (Ursidae Body, Lacertilia Head, Felidae Head, Testudines Body, Testudines Head), respectively. For *sys* (e), the testing combination is (Snake Head, Car Tier, Felidae Head, Bird Head, Bovidae Body), in which these concepts are all seen in *continual*. For *pro* (f), the testing combination is (Bovidae Foot, Ursidae Body, Bird Foot, Ursidae Head, Lacertilia Head, Bird Head, Testudines Head). For *non* (g), the trained combination (Ursidae Body, Lacertilia Head, Felidae Head, Testudines Body, Testudines Head) is tested. For *noc* (h), (Bicycle Body, Aeroplane Head, Bicycle Seat, Aeroplane Tail, Boat Sail) contains unseen concepts.

Since the concepts (e.g., *Bird*) in CGQA are somewhat coarse-grained and the maximally allowed number of concepts in one combination is $4$ (i.e., $2 \times 2$), one may be interested in the performance of a continual learner on finer-grained datasets (e.g., *Bird Wings*) and more concepts in one combination (e.g., $3 \times 3$). One alternative benchmark is Compositional PartImageNet (CPIN) constructed from the PartImageNet [9] dataset with a similar process as CGQA. PartImageNet is a subset of ImageNet [5] and provides segmentation for parts of objects (e.g., wings of birds), which can be used to construct fine-grained datasets. However, PartImageNet only provides bounding boxes but no other meta-information for concepts. Thus, we do not construct the *sub* testing scheme, since it needs the attribute information of concepts. Further, instead of constructing an image with $2 \times 2$ possible positions for concepts, we allow a combination of maximally $3 \times 3$ concepts. The tasks constructed from the CPIN benchmark are generally more difficult than those constructed from the CGQA benchmark. We give examples of images in Figure 3.

### C.2    Compositional Objects365 (COBJ) Benchmark

The source dataset is Objects365 [34] (License CC BY 4.0). Objects365 is a high-resolution dataset designed for object detection with enough combinations of objects in the wild.

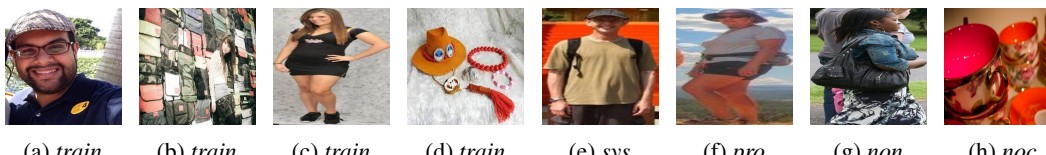

| (a) *train* | (b) *train* | (c) *train* | (d) *train* | (e) *sys* | (f) *pro* | (g) *non* | (h) *noc* |

Figure 4: Image examples for COBJ. The labels for *continual*, i.e., *train* (a-d) are (Glasses, Hat, Person), (Handbag/Satchel, Person), (Bracelet, Person, Sneakers), and (Bracelet, Hat), respectively. For *sys* (e), the testing combination is (Bracelet, Hat, Person), in which grass and shirt are all seen in *continual*. For *pro* (f), the testing combination is (Glasses, Handbag/Satchel, Hat, Person, Sneakers). For *non* (g), the trained combination (Handbag/Satchel, Person) is tested. For *noc* (h), (Cup, Plate) contains unseen cup and plate concepts.

### C.2.1 Construction of COBJ

Our constructed COBJ benchmark consists of a continual training phase and four few-shot testing phases, i.e., *sys*, *pro*, *non*, and *noc*. Note that the original Objects365 benchmark does not provide attribute information. As a result, we do not construct *sub* testing phase. Figure 4. We first pick $M = 20$ objects that have as many combinations with others and as many instances as possible and larger than 30 pixels on width and height utilizing the provided bounding boxes. These objects form our concept pool $\mathcal{C}$ in the continual training phase. After that, we collect images in the Objects365 benchmark that contains the selected objects. Next, we assign the combination of objects (concepts) in one pre-processed image as its label. Note that, although Objects365 provides 365 objects with bounding boxes, most of them are long-tailed. Thus, it is not guaranteed for the learners to learn the compositionality according to the sample efficiency experiment and the analysis in section E.6. It is worth noting that, we still utilize these objects to construct our *noc* set to test the concept-level generalization capability. Then, we split the image instances into *train*, *val*, *test*, and *fewshot* sets and make sure the learner will not see exactly the same instance in different sets. Among these sets, *train*, *val*, and *test* are used in the continual training phase, and *fewshot* is used in the few-shot testing phases.

### C.2.2 Continual Training Phase

The *continual* training set $\mathcal{Y}_{tr}$ contains 30 different object combinations and each $y_i \in \mathcal{Y}_{tr}$ consists of $M_{tr} = 3$ (or 2) objects in $\mathcal{C}$, as shown in Figure 4a-4d. We select around 1000 (*train*), 50 (*val*), 100 (*test*) image instances for each combination from the corresponding *train*, *val*, *test* sets, respectively. Generally, we construct $T = 3$ tasks, each with $N = 10$ object combinations $\{y_i\}_{i=1}^N$. We also explore different $T$ in section E.5 to examine the effect of the number of classes in one task.

### C.2.3 Few-shot Testing Phase

In this phase, we have four different testing schemes, i.e., *sys*, *pro*, *non*, and *noc*. Note that we do not construct *sub* since their is no attribute label for the image. We will introduce all these schemes in detail as follows:

1. **Systematicity Novel Testing (*sys*)** We select another set $\mathcal{Y}_{sys}$ with 30 novel combinations different from $\mathcal{Y}_{tr}$. The example images are shown in Figure 4e. Different from the large *train* set, we only need to generate 100 image instances for each combination from the *fewshot* set. For clarity, we also name this image set as *sys*.
2. **Productivity Novel Testing (*pro*)** In the *pro* test, more concepts are in one image than that in the continual training phase. Specifically as shown in Figure 4f, 4 or more objects (concepts) are contained in one image. We sample 100 image instances for each combination from the *fewshot* set and name this image set as *pro*.
3. **Non-novel Testing (*non*)** Different from the *continual* training phase in which training objects are from the *train* set, in the *non* test, we sample object instances from the *fewshot* set to construct 100 image instances and name this image set as *non*. The example images are shown in Figure 4g.
4. **Non-compositional Testing (*noc*)** We construct another novel concept pool $\mathcal{C}_{noc}$, with other 19 object combinations in Objects365 that contain the unseen objects rather than the seen objects

in $\mathcal{C}_{tr}$. Further, we design 100 object combinations $\mathcal{Y}_{noc}$ and each with 100 image instances and name this image set as *noc*. The example images are shown in Figure 4h.

From the obtained image sets (i.e., *sys*, *pro*, *non*, and *noc*), we construct $T_{nv}$ $N$-way $K$-shot tasks for each set by randomly selecting images, and $T_{nv} = 300, N = 10, K = 10$. The reported average test accuracy is used as the metric to analyze the compositional ability of the learner after continual training. We also explore different $N$ in section E.5 to examine the effect of the number of classes in one task.

### C.2.4 Training Order

Unless stated otherwise, the 20 training concepts for COBJ are: *Basketball, Bench, Boat, Bracelet, Car, Chair, Desk, Glasses, Guitar, Handbag/Satchel, Hat, Lamp, Microphone, Necklace, Other Shoes, Person, SUV, Sailboat, Sneakers, Street Lights*. We choose these 20 concepts because they have numerous combinations and instances in the original dataset. The combinations of concepts need to be sufficiently explored among the 3 tasks. The continual training order can be generated randomly. For the sake of analysis, we fix the continual training order in the experiments. The following shows the order:

- Task 1: *(Bracelet, Person, Sneakers), (Car, SUV, Street Lights), (Bracelet, Glasses, Person), (Bench, Person), (Person, Sneakers), (Car, Person), (Boat, Sailboat), (Lamp, Person), (Glasses, Hat, Person), (Chair, Person)*;
- Task 2: *(Hat, Person, Sneakers), (Microphone, Person), (Basketball, Person, Sneakers), (Boat, Person), (Other Shoes, Person, Sneakers), (Other Shoes, Person), (Guitar, Microphone, Person), (Bracelet, Necklace, Person), (Glasses, Person), (Necklace, Person)*;
- Task 3: *(Car, Street Lights), (Car, Person, Street Lights), (Glasses, Person, Sneakers), (Person, Street Lights), (Hat, Other Shoes, Person), (Chair, Desk), (Hat, Person), (Bracelet, Person), (Handbag/Satchel, Person), (Glasses, Necklace, Person)*.

## D Training Details

By default, we resize the input samples to $128 \times 128$ pixels in RGB color channels and perform the random horizontal flipping and normalization with ImageNet statistics when training a ResNet-18 model. For inference, we remove the random horizontal flipping part for consistency. Additionally for the training of ViT backbones, we use an image size of $224 \times 224$ and add Rand-Augment [4] and random erasing [45] for training to further improve the results. We also use a cosine learning rate schedule with 200 training epochs and an early stop strategy with a patient of 5 epochs using *val* sets. The ViT structure we use is as follows: patch size 16*16, hidden dimension 384, number of layers 9, multi-head attention with 16 heads and feed-forward with dimension 1536. We adopt Avalanche [20] library based on Pytorch for our experiments. All experiments are executed on a single NVIDIA TESLA V100 GPU.

### D.1 Hyper-parameters (CGQA)

In this section, we report the hyper-parameters tuning grids in the continual training phase. For the few-shot testing phases, the learning rate is fixed at 1e-3. Unshown hyper-parameters use their default values. Unless stated otherwise, the **bold** value is chosen in our experiments for specific methods. Note that, methods using multi-head classifiers are denoted with a postfix "*" (e.g., Finetune*).

- ResNet-18 backbone
  learning rate: [1e-4, 2e-4, 3e-4, **4e-4** (MNTDP*), 5e-4, **8e-4** (ER*), **1e-3** (GEM*, RPSnet, MT*), **3e-3** (Finetune, ER), **5e-3** (LwF, EWC, EWC*, MT), **8e-3** (Finetune*), **1e-2** (GEM, LwF*), 3e-2, 5e-2, 8e-2, 0.1]
  memory size: [**1000** (ER, ER*, RPSnet)]
  patterns per exp: [**32** (GEM, GEM*), 64, 128, 256]
  mem strength: [0.1, 0.2, **0.3** (GEM, GEM*), 0.4, 0.5]
  alpha: [0.1, 0.5, **1** (LwF, LwF*), 5, 10]
  temperature: [0.1, 0.5, **1** (LwF, LwF*), 2]
  lambda: [**0.1** (EWC), 0.5, 1, 1.5, 2 (EWC*)]
- ViT backbone

Table 1: Results on CGQA with ResNet-18 backbone. Accuracy $(\%) \pm 95\%$ confidence intervals $(\%)$ over 300 tasks are reported for few-shot testing phases. Task-IL settings denote with a postfix "*".

| Methods | $A_{con}$ | $A_{sys}$ | $A_{pro}$ | $A_{sub}$ | $H_n$ | $A_{non}$ | $A_{noc}$ | $H_r$ | $H_a$ |
|---|---|---|---|---|---|---|---|---|---|
| MultiTask | 83.61 | $88.14 \pm 0.62$ | $85.94 \pm 0.65$ | $69.67 \pm 0.75$ | 80.35 | $91.55 \pm 0.51$ | $40.04 \pm 0.99$ | 55.71 | 68.28 |
| Finetune | 8.38 | $64.73 \pm 0.78$ | $65.43 \pm 0.73$ | $61.26 \pm 0.67$ | 63.75 | $68.54 \pm 0.80$ | $40.32 \pm 0.72$ | 50.77 | 57.84 |
| ER | 19.78 | $\mathbf{71.38 \pm 0.75}$ | $70.11 \pm 0.64$ | $64.32 \pm 0.69$ | 68.46 | $\mathbf{77.27 \pm 0.67}$ | $40.98 \pm 0.72$ | 53.56 | 61.60 |
| GEM | 8.56 | $66.56 \pm 0.77$ | $\mathbf{73.47 \pm 0.61}$ | $62.80 \pm 0.71$ | 67.33 | $72.65 \pm 0.73$ | $41.64 \pm 0.72$ | 52.94 | 60.72 |
| LwF | 9.11 | $71.22 \pm 0.74$ | $73.28 \pm 0.61$ | $\mathbf{68.74 \pm 0.66}$ | $\mathbf{71.03}$ | $76.56 \pm 0.66$ | $\mathbf{48.69 \pm 0.77}$ | $\mathbf{59.52}$ | $\mathbf{65.93}$ |
| EWC | 8.22 | $64.99 \pm 0.78$ | $\mathbf{73.47 \pm 0.64}$ | $63.25 \pm 0.66$ | 66.95 | $69.03 \pm 0.74$ | $41.38 \pm 0.71$ | 51.75 | 59.91 |
| RPSnet | $\mathbf{33.45}$ | $59.80 \pm 0.83$ | $60.26 \pm 0.72$ | $59.75 \pm 0.74$ | 59.94 | $64.22 \pm 0.81$ | $45.09 \pm 0.64$ | 52.98 | 56.95 |
| MultiTask* | 92.17 | $82.16 \pm 0.68$ | $84.22 \pm 0.60$ | $71.82 \pm 0.75$ | 79.01 | $86.44 \pm 0.69$ | $44.07 \pm 0.95$ | 58.38 | 69.23 |
| Finetune* | 72.46 | $70.32 \pm 0.73$ | $72.62 \pm 0.63$ | $66.33 \pm 0.69$ | 69.66 | $75.32 \pm 0.70$ | $43.26 \pm 0.73$ | 54.95 | 62.92 |
| ER* | 76.05 | $\mathbf{71.37 \pm 0.70}$ | $72.67 \pm 0.69$ | $\mathbf{66.80 \pm 0.63}$ | $\mathbf{70.19}$ | $\mathbf{76.28 \pm 0.66}$ | $45.61 \pm 0.77$ | $\mathbf{57.09}$ | $\mathbf{64.29}$ |
| GEM* | 21.60 | $69.44 \pm 0.75$ | $73.14 \pm 0.67$ | $66.61 \pm 0.66$ | 69.63 | $73.31 \pm 0.71$ | $\mathbf{45.97 \pm 0.74}$ | 56.51 | 63.71 |
| LwF* | 73.19 | $69.61 \pm 0.77$ | $\mathbf{74.22 \pm 0.63}$ | $65.00 \pm 0.65$ | 69.40 | $75.25 \pm 0.69$ | $42.52 \pm 0.72$ | 54.34 | 62.48 |
| EWC* | 71.10 | $69.40 \pm 0.79$ | $72.37 \pm 0.65$ | $65.10 \pm 0.67$ | 68.83 | $74.99 \pm 0.72$ | $42.82 \pm 0.77$ | 54.51 | 62.29 |
| MNTDP* | 68.98 | $47.27 \pm 0.91$ | $47.41 \pm 0.86$ | $47.50 \pm 0.85$ | 47.39 | $52.28 \pm 0.81$ | $32.29 \pm 0.83$ | 39.93 | 44.09 |

learning rate: [**1e-5** (GEM*), **5e-5** (GEM, RPSnet), **1e-4** (MT, MT*, Finetune, Finetune*, ER, ER*, LwF, LwF*, EWC, EWC*, MNTDP), 5e-4, 1e-3]
memory size: [**2000** (ER, ER*, RPSnet)]
patterns per exp: [**32** (GEM, GEM*), 64, 128, 256]
mem strength: [0.1, 0.2, **0.3** (GEM, GEM*), 0.4, 0.5]
alpha: [0.1, 0.5, **1** (LwF, LwF*), 5, 10]
temperature: [0.1, 0.5, **1** (LwF, LwF*), 2]
lambda: [**0.1** (EWC), 0.5, 1, 1.5, 2 (EWC*)]

## D.2 Hyper-parameters (COBJ)

- ResNet-18 backbone
  learning rate: [1e-4, **4e-4** (MNTDP) **5.3e-4** (EWC), **1e-3** (MT, MT*, Finetune, Finetune*, GEM, GEM*, LwF, LwF*, RPSnet), **1e-2** (ER, ER*, EWC*), 1e-1]
  memory size: [**1000** (ER, ER*, RPSnet)]
  patterns per exp: [**16** (GEM*), 32, 64, 128, **256** (GEM)]
  mem strength: [**0.00139** (GEM), 0.1, 0.2, **0.3** (GEM*), 0.4, 0.5]
  alpha: [0.01, 0.1, **1** (LwF, LwF*), 10, 100]
  temperature: [0.01, 0.1, 0.5, 1, **1.52** (LwF), **2** (LwF*), 10]
  lambda: [1e-4, 1e-3, 1e-2, 0.1, 1, **10** (EWC), **100** (EWC*)]
- ViT backbone
  learning rate: [**1e-5** (RPSnet), **5e-5** (MT, MT*, Finetune, Finetune*, ER, ER*, GEM*, LwF*), **1e-4** (GEM, LwF, EWC, EWC*), **5e-4** (MNTDP*), 1e-3]
  memory size: [**2000** (ER, ER*, RPSnet)]
  patterns per exp: [**32** (GEM, GEM*), 64, 128, 256]
  mem strength: [0.1, 0.2, **0.3** (GEM, GEM*), 0.4, 0.5]
  alpha: [0.1, 0.5, **1** (LwF, LwF*), 5, 10]
  temperature: [0.1, 0.5, **1** (LwF, LwF*), 2]
  lambda: [0.1, 0.5, 1, 1.5, 2 (EWC, EWC*)]

# E Additional Results

In this section, we present a comprehensive analysis as the supplementation of the results in the main text.

## E.1 CGQA Overall Results

The reported results on CGQA are shown in Figure 1, which are the same as in the main text. On the class-IL setting, methods equipped with the memory buffer (i.e., ER, RPSnet) outperform the others in the continual training phase. RPSnet wins all baselines in the *continual* training phase since it trains one specific path of modules for one task to overcome forgetting. However, in the

few-shot novel-testing phases (i.e., *sys, pro, sub*), RPSnet shows no superiority compared with other approaches. This indicates the reused knowledge is not compositional enough to handle the novel recombination of seen concepts. LwF beats others instead (according to the $H_n$ result), although its *continual* performance is not outstanding. This indicates that the learned feature extractor by LwF is excellent in terms of extracting compositional features but the classifier fails to give a correct prediction. We further discuss this prediction bias phenomenon with CAM visualization in section E.3. Regularization-based mechanisms, such as GEM, LwF, and EWC can hardly solve this issue in the classifier. As a result, only replay-based methods (e.g., ER, RPSnet) have improved performance. To provide more insights beyond average testing accuracy for *continual*, we resort to the few-shot testing to test feature extractors' performance instead. The good performance of MultiTask on few-shot testing and CAM visualization shows its superior compositionality.

On the task-IL setting, MNTDP* does not perform well in both the continual training and the few-shot testing phases. This is because forgetting is not as suffered as in the class-IL setting. Thus, simply freezing modules does not provide efficient reusable (compositional) knowledge for the novel recombination of seen concepts. The above results on class-IL and task-IL show that the modularity-based approaches are not necessarily better at extracting compositional features from the input images. Compared with MultiTask, the forgetting also reflects on the degradation of compositionality, since no method can outperform MultiTask in the few-shot testing phase.

**Detailed analysis on few-shot tests**  For each method, $A_{non} > A_{pro} > A_{sys} > A_{sub} > A_{noc}$ in general. The performances on *non* are the best, which is as expected since models prefer the seen combinations and suffer compositional gaps in novel combinations to some extent. This performance gap is acceptable for *sys* and *pro* but large for *sub* on the contrary. This is because *sub* needs to compose attributes of concepts to form unseen recombination which needs the higher-level compositional ability than *sys* and *pro*. Interestingly, we observe contrary results when models are with ViT backbone (i.e., $A_{sub} > A_{pro} > A_{sys}$), which is detailed in section E.8. This is because the selected attributes for concepts in *sub* are mainly texture attributes (e.g., color). This observation meets the analysis in [29] that the multi-head attention layer has a bias on shape attributes while the convolution layer has a bias on texture attributes. Thus, models with ResNet-18 backbone are more sensitive to the texture difference between *continual* and *sub*. The accuracy on *pro* is generally slightly better than that on *sys*, except on MultiTask and ER, showing that having more concepts can, to some extent, help to understand the images. The generally lower accuracy on the single-head baselines (class-IL continual training setting) compared with their multi-head counterpart (task-IL continual training setting), except LwF, shows that the single-head classifier makes it difficult for the feature extractor to remember the compositional knowledge.

**Modularity-based methods**  Further, RPSnet has the worst performance on $H_n$ and $A_{non}$. This counter-intuitive observation hints that the learned feature extractor is not wise enough. Its best performance on *continual* relates to the memory buffer (which reduces the prediction bias in the classifier) and the large model capacity for inference. Although RPSnet does not have outstanding compositional generalization capability, it still has the second best *noc* result (LwF is the best), showing that it has an advantage in handling generalization on unseen concepts. Not surprisingly for MNTDP*, the few-shot testing cases are the worst in the task-IL setting. This is because the learned reusable knowledge is not compositional, such that it can not be sufficiently used for the cases of novel recombination.

## E.2  COBJ Overall Results

In this subsection, we study methods on COBJ under different numbers of classes in each task. Since the total number of classes in COBJ is 30, we investigate three settings: ten 3-way tasks, five 6-way tasks, and three 10-way tasks. The experimental results are shown in Table 2, Table 3, and Table 4. The main observations are similar to that on CGQA as we discuss before. It is surprising that RPSnet works the best in the class-IL setting, which indicates its superiority when dealing with real-world tasks. However, MNTDP* still shows no outstanding compositional generalization capability in the task-IL setting, although its continual testing accuracy is the best when the number of training tasks is large (i.e., ten 3-way tasks and five 6-way tasks). Modularity-based methods freeze the modules for old tasks that may not be suitable for new tasks unless the modules are compositional enough.

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

introduced by the continual training process and non-i.i.d task distribution are the main cause of the poor continual testing performance.

## E.4 Old and Fresh Concepts

This subsection is the supplementation of the **Concept-level forgetting** in the main text. We are going to investigate the forgetting performance w.r.t. concepts. That is to answer the following question: how is the few-shot testing performance on two task sets consisting of freshly learned concepts and old concepts, respectively?

We first divide the concepts in CGQA into two groups (i.e., **old** and **fresh**) according to whether they are seen in Task 10 (the last continual training task). According to the training order shown in section C.1.4, the **fresh** group contains 14 concepts: *(building, car, door, fence, flower, hat, helmet, leaves, pants, pole, shirt, shorts, sign, wall)*, in which the knowledge of these concepts is updated in Task 10. The remaining seven concepts *(bench, chair, grass, jacket, plate, shoe, table)* are in the **old** group, since they may suffer forgetting. Interestingly, we also find a low-frequent concept *table* which is only seen fourth among all 10 tasks. After that, we collect specific combinations for few-shot testing in the following:

- *non(f)* (the subset of *non* with only **fresh** concepts): *(building, hat), (building, leaves), (car, flower), (car, helmet), (car, leaves), (car, pants), (car, shirt), (car, shorts), (car, sign), (door, fence), (door, leaves), (door, pole), (fence, flower), (fence, helmet), (fence, leaves), (fence, pants), (fence, pole), (fence, shorts), (fence, wall), (flower, helmet), (flower, pants), (flower, pole), (flower, shirt), (flower, wall), (hat, shirt), (hat, shorts), (hat, sign), (hat, wall), (helmet, leaves), (helmet, pole), (helmet, shirt), (helmet, shorts), (helmet, sign), (leaves, shirt), (leaves, shorts), (leaves, sign), (leaves, wall), (pants, shirt), (pants, shorts), (pants, wall), (pole, shirt), (pole, wall), (shirt, wall).*
- *non(o)* (the subset of *non* with only **old** concepts): *(bench, chair), (bench, jacket), (bench, table), (chair, grass), (chair, shoe), (grass, jacket), (grass, plate), (grass, shoe), (grass, table), (jacket, shoe).*
- *sys(f)* (the subset of *sys* with only **fresh** concepts): *(building, car), (building, door), (building, fence), (building, helmet), (building, pants), (building, shirt), (building, shorts), (building, wall), (car, door), (car, fence), (car, hat), (car, pole), (car, wall), (door, flower), (door, hat), (door, helmet), (door, pants), (door, shirt), (door, shorts), (door, sign), (door, wall), (fence, hat), (fence, shirt), (fence, sign), (flower, hat), (flower, leaves), (flower, shorts), (flower, sign), (hat, helmet), (hat, leaves), (hat, pants), (helmet, pants), (leaves, pants), (leaves, pole), (pants, pole), (pants, sign), (pole, shorts), (pole, sign), (shirt, shorts), (shirt, sign), (shorts, wall), (sign, wall).*

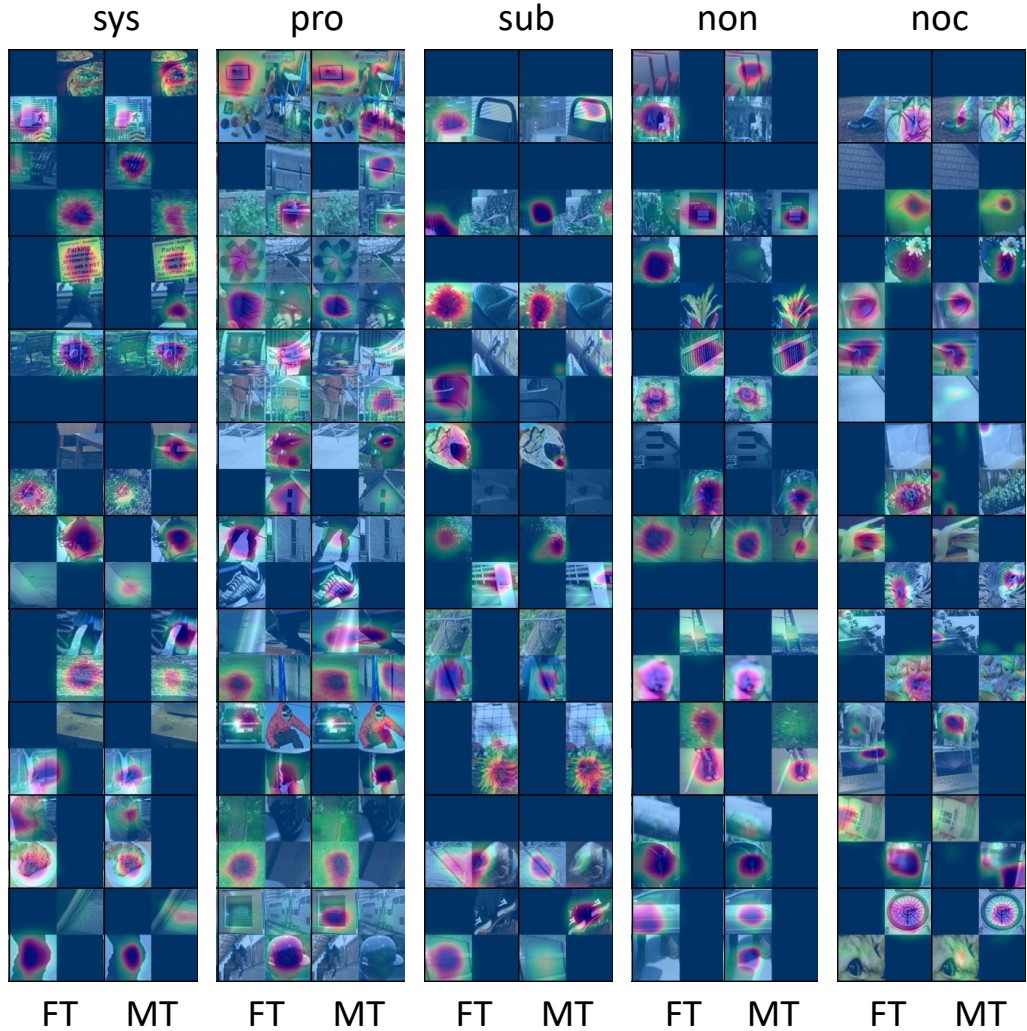

Figure 5: CAM visualization for MultiTask (MT) and Finetune (FT) on one sampled CGQA task for each fewshot testing. **Take away**: In general, FT works well on recognizing concepts, while it is still inferior to MT.

- ***sys(o)*** (the subset of *sys* with only **old** concepts): *(bench, grass), (bench, plate), (bench, shoe), (chair, jacket), (chair, plate), (chair, table), (jacket, plate), (jacket, table), (plate, shoe), (plate, table), (shoe, table).*

We generate 300 2-way 10-shot tasks for each subset to evaluate the models after the continual training.

The average test accuracy and the related score are shown in Figure 7. Note that we calculate the related score $(S(f), S(o))$ for fresh and old tasks, separately. For example, $S(f) = (A_{sys(f)} - A_{non(f)})/A_{non(f)}$, which is the performance gap between *sys(f)* and *non(f)*. A large value of the related score indicates a small forgetting of the specific concepts. We can observe that the $A_{non(f)} > A_{non(o)}, A_{sys(f)} > A_{sys(o)}$, which are as expected since fresh concepts are newly updated on the last continual task. And $S(f) > S(o)$, which indicates that the learner has better systematicity compositionality on fresh than on old concepts. The above observation shows that the learner indeed learns the composition concept-wisely and the forgetting indeed shows in the old concepts. On the other hand, MultiTask, which is regarded as the upper bound, suffers the smallest performance drop on both fresh and old tasks. It is as expected since the learner jointly

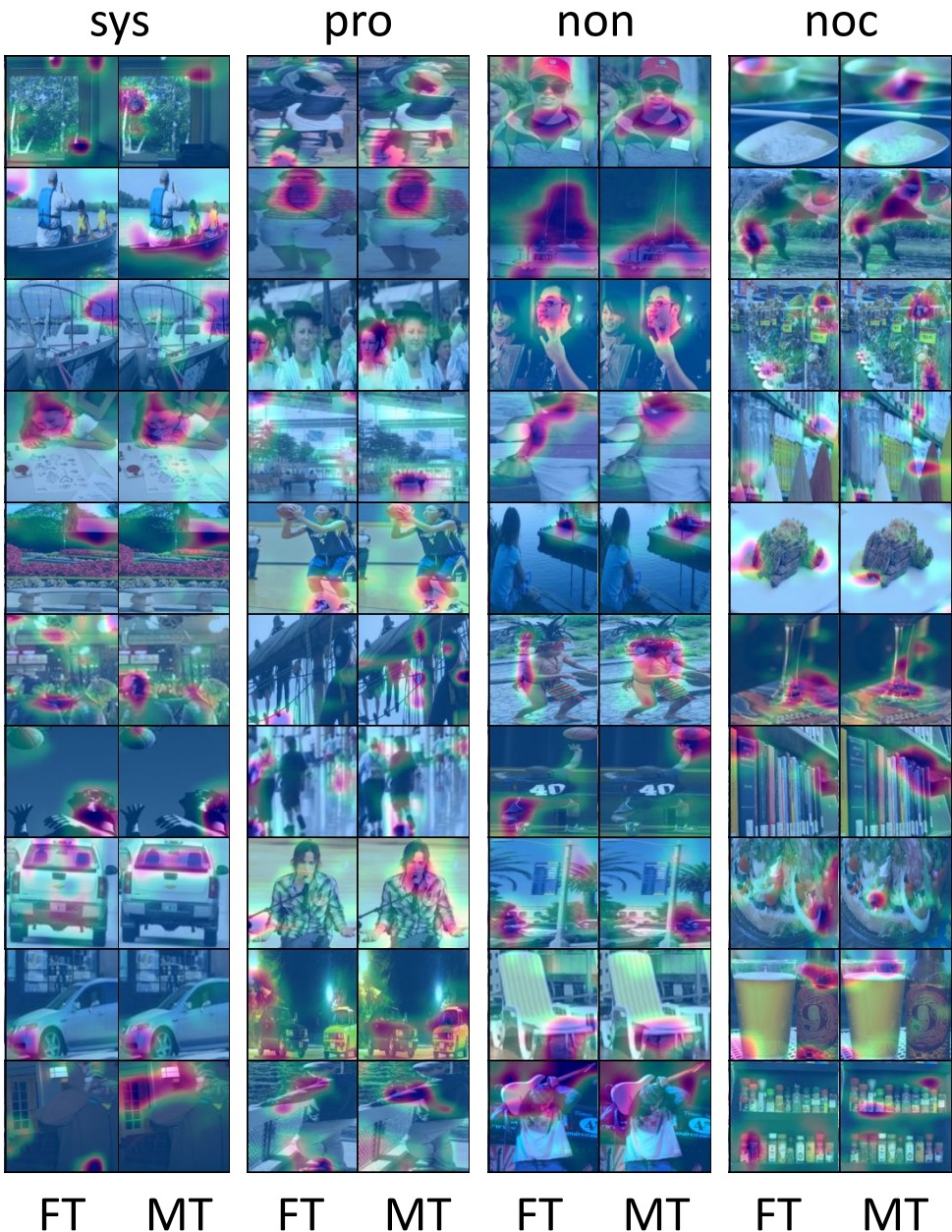

Figure 6: CAM visualization for MultiTask (MT) and Finetune (FT) on one sampled COBJ task for each fewshot testing. **Take away**: In general, FT works well on recognizing concepts, while it is still inferior to MT.

learns all continual tasks. This observation also indicates that the continual learner suffers forgetting not only on old tasks but also on old concepts.

We also use CAM [46] to visualize the class activation region on the image, taking Finetune and MultiTask as examples in Figure 8. We randomly select five samples for each class. It is obscure to discriminate each case (e.g., *non(f)*) with CAM visualization. However, our few-shot testing method can clearly and statistically show the different compositional generalization capacities via testing accuracy which indicates the superiority.

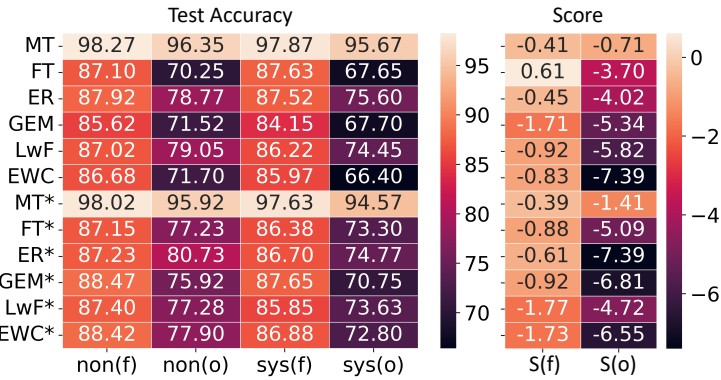

| | Test Accuracy | | | | Score | |
|---|---|---|---|---|---|---|
| | non(f) | non(o) | sys(f) | sys(o) | S(f) | S(o) |
| MT | 98.27 | 96.35 | 97.87 | 95.67 | -0.41 | -0.71 |
| FT | 87.10 | 70.25 | 87.63 | 67.65 | 0.61 | -3.70 |
| ER | 87.92 | 78.77 | 87.52 | 75.60 | -0.45 | -4.02 |
| GEM | 85.62 | 71.52 | 84.15 | 67.70 | -1.71 | -5.34 |
| LwF | 87.02 | 79.05 | 86.22 | 74.45 | -0.92 | -5.82 |
| EWC | 86.68 | 71.70 | 85.97 | 66.40 | -0.83 | -7.39 |
| MT* | 98.02 | 95.92 | 97.63 | 94.57 | -0.39 | -1.41 |
| FT* | 87.15 | 77.23 | 86.38 | 73.30 | -0.88 | -5.09 |
| ER* | 87.23 | 80.73 | 86.70 | 74.77 | -0.61 | -7.39 |
| GEM* | 88.47 | 75.92 | 87.65 | 70.75 | -0.92 | -6.81 |
| LwF* | 87.40 | 77.28 | 85.85 | 73.63 | -1.77 | -4.72 |
| EWC* | 88.42 | 77.90 | 86.88 | 72.80 | -1.73 | -6.55 |

Figure 7: Fresh tasks *vs* old tasks on CGQA. We also report the related score for sys(f) and sys(o) w.r.t. non(f) and non(o), respectively. MT stands for MultiTask and FT stands for Finetune. **Take away**: fresh concepts suffer little performance drop when testing on novel combinations (sys(f)) while old concepts on the contrary.

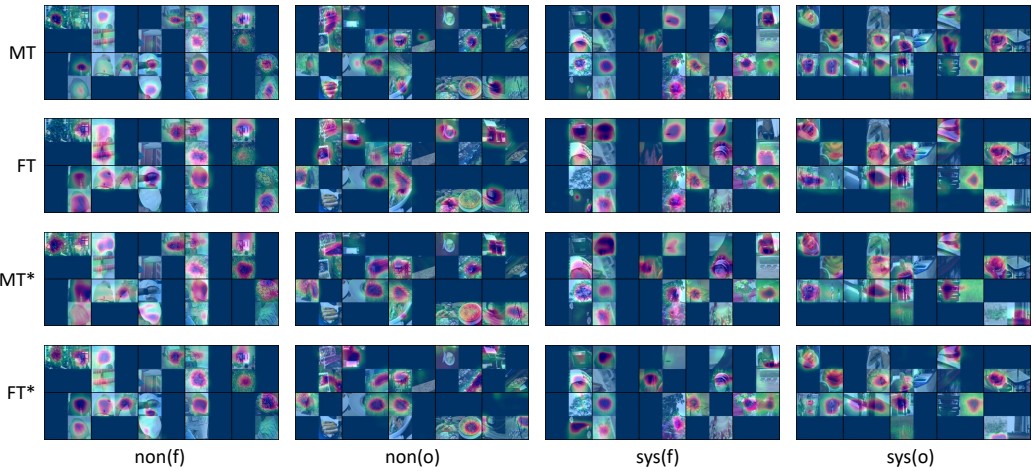

Figure 8: CAM visualization for MultiTask (MT) and Finetune (FT) on four task sets. **Take away**: Results are consistent with the test accuracy results in Figure 7. We can roughly observe that Finetune can locate well on both concepts in non(f) but tend to only find one concept in others.

### E.5 Effect of Number of Classes in Continual Tasks

In this subsection, we further investigate the effect of the number of training tasks on the compositional generalization. When the number of training tasks increases, the number of classes in each task decreases since the total number of classes is constant. Thus, the learner needs a smaller number of compositional features necessary for distinguishing classes in each task, comparing with one task containing all classes. For example, one can distinguish *horse* with *person* by their different shapes. But this is not enough for the case of *horse* and *zebra* (i.e., limited compositionality). However, for the tri-classification task of distinguishing among *horse*, *zebra*, and *person*, one can learn both shape and texture features (i.e., relatively better compositionality). After training the learners on three (10-way), five (6-way), and ten (3-way) COBJ tasks, respectively, we evaluate them on few-shot *sys* and *pro* tasks and visualize $H_n$. The number of classes in these few-shot tasks is ten (denoted as 10way-$H_n$), six (denoted as 6way-$H_n$), and three (denoted as 3way-$H_n$), respectively. The observation in Figure 9 meets the above intuitive understanding. The more training tasks (thus the number of classes in each task is shorter), the smaller $H_n$ nearly for all methods. We can also observe that this performance drop is consistent with different numbers of classes in the few-shot tasks (left/right column in Figure 9). Although on CGQA, RPSnet shows poor compositional generalization capability, it is the best continual learning algorithm on COBJ. While as for MNTDP*,

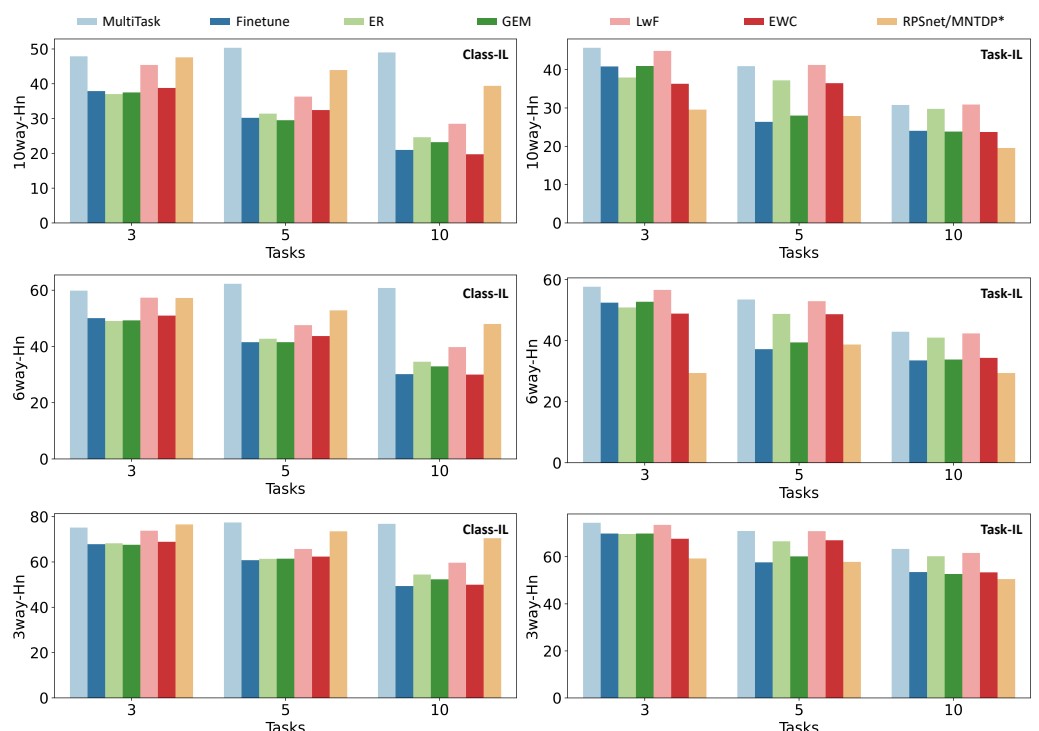

Figure 9: Number of continual tasks *vs* Hn on COBJ testing tasks. The top row is methods tested with 10-way tasks, while the middle row is methods tested with 6-way tasks, and the bottom row is methods tested with 3-way tasks. **Take away**: A larger number of continual tasks (smaller ways for each task) hinders the compositional generalization capability. The conclusion holds for different numbers of classes in the few-shot tasks.

it still does not show superior compositional generalization capability, indicating that large efforts are still needed on modularity-based methods for not bringing benefits to compositional generalization. The performance gaps on $H_n$ between MultiTask and other continual learning methods show that compositional generalization capability is still a big question for the continual learning community to answer.

### E.6 Sample Efficiency for Learning Compositionality

To examine how many training samples are needed for a continual learner to learn the compositional generalization ability, we vary the number of training samples for each class in the CGQA continual training tasks. Intuitively, a small number of training samples hinders not only the cases of novel combinations (i.e., *sys, pro, sub*) but also the case of the seen combination (i.e., *non*), as shown in the left column of Figure 10. Instead, we resort to the related score (i.e., $S(sys), S(pro), S(sub)$, and $S(noc)$) between each few-shot testing result with the *non* testing result. For example, $S(sys) = (A_{sys} - A_{non})/A_{non}$, where a larger positive $S(sys)$ indicates a better compositional generalization ability on Systematicity and a smaller negative $S(sys)$ shows that the learner does not well-handle the novel combinations of seen concepts. The related score results are visualized in the right column of Figure 10. For all the testing cases (i.e., *sys, pro, sub, noc*), the related scores converge if trained with more than 300 samples for each class. After this threshold, the ratio of the improvement for each few-shot test to *non* is unchanged. We can conclude that 300 samples for each class are needed for learners to learn compositionality on CGQA.

### E.7 Effect of Frozen Feature Extractor

Without freezing the feature extractor, the learner needs to update more parameters (i.e., the feature extractor and the classifier) from only a small amount of support samples in the few-shot tasks. As

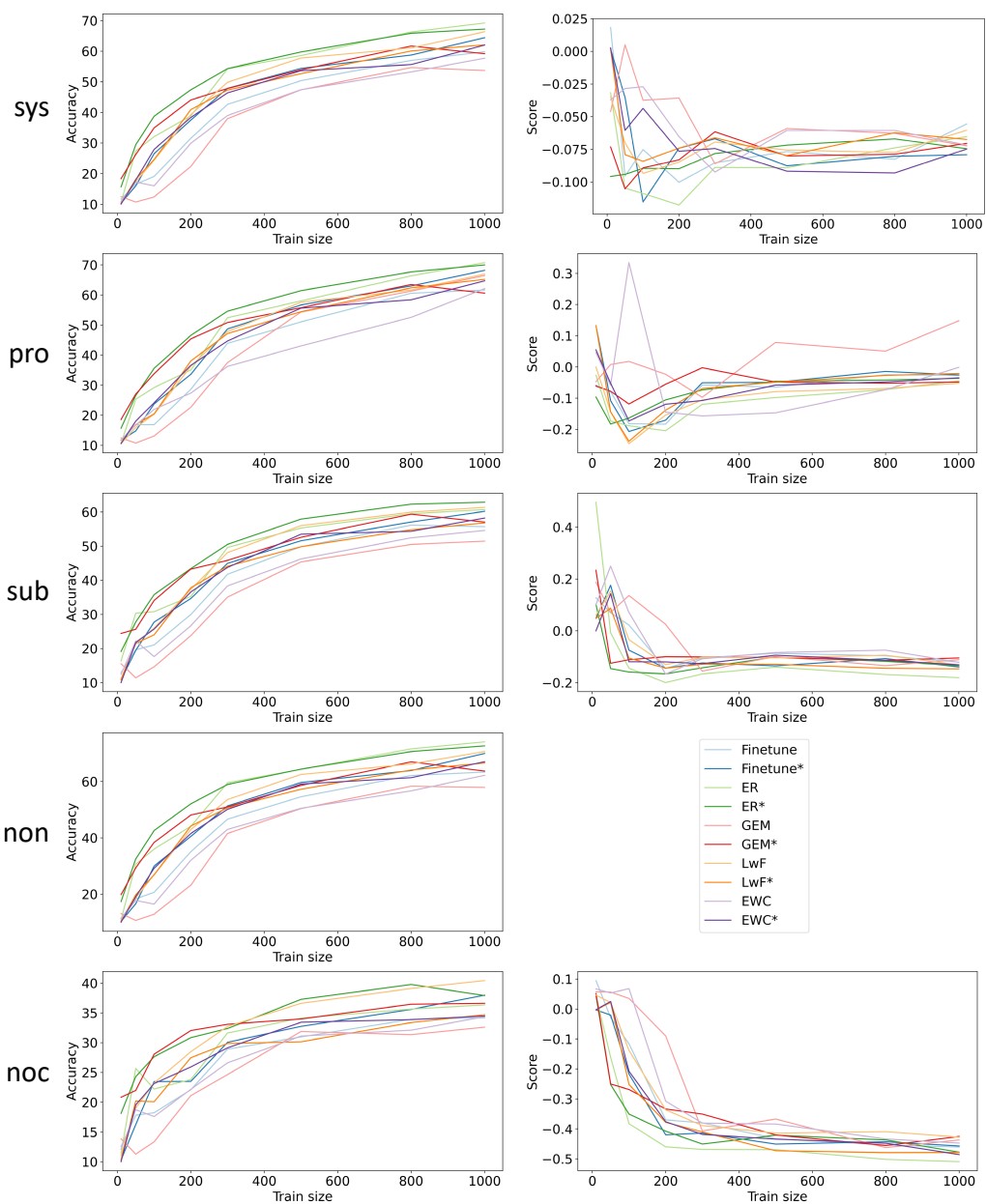

Figure 10: Varying training size on CGQA. Scores denote related scores for *sys, pro, sub, noc* w.r.t. *non*. **Take away**: Accuracy is improved with more training samples for all fewshot testing cases. Related scores become smooth and unchanged if more than 300 training samples for each class. This fact shows that 300 samples are needed for learning compositionality.

a result, it will suffer from severe overfitting. We evaluate the learners obtained by five baselines after the continual training phase with and without (marked with a postfix "†") freezing the feature extractor in Figure 11. We observe that methods that freeze the feature extractor generally have better testing accuracy on few-shot testing schemes than without. This observation supports the above claim. Interestingly, we find that ER† nearly fails in all cases (i.e., with an accuracy smaller than 20% on 10-way tasks), although it has the best average test accuracy in the continual training phase. Remind that the goal of few-shot testing schemes is to evaluate whether the continually trained feature extractor is able to extract features corresponding to our expected concepts. Thus, we do not allow the feature extractor to learn from the support samples in the few-shot tasks. Unless otherwise stated, we freeze the feature extractor of a continual learner in the few-shot testing phases.

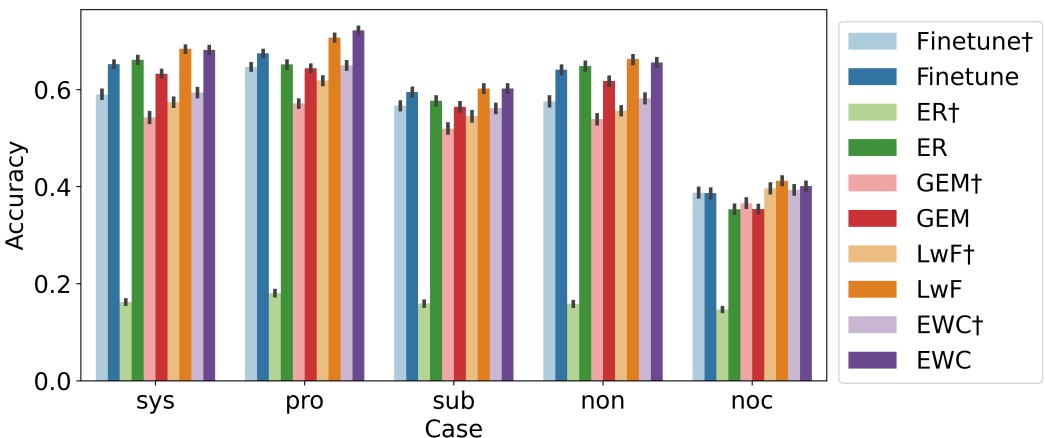

Figure 11: Histogram of test accuracy on CGQA. Methods without freezing the feature extractors when training on few-shot tasks are denoted with a postfix "†". **Take away**: methods without freezing the feature extractor will generally obtain poorer few-shot testing performance and may encounter severe overfitting, especially in ER.

Table 5: Results on CGQA (10-way) with ViT backbone. Accuracy $(\%) \pm 95\%$ confidence intervals $(\%)$ over 300 tasks are reported for few-shot testing phases. Task-IL settings denote with a postfix "*".

| Methods | $A_{con}$ | $A_{sys}$ | $A_{pro}$ | $A_{sub}$ | $H_n$ | $A_{non}$ | $A_{noc}$ | $H_r$ | $H_a$ |
|---|---|---|---|---|---|---|---|---|---|
| MultiTask | 66.00 | $83.68 \pm 0.60$ | $87.77 \pm 0.48$ | $80.53 \pm 0.56$ | 83.89 | $89.99 \pm 0.49$ | $56.28 \pm 0.75$ | 69.25 | 77.35 |
| Finetune | 8.59 | $50.49 \pm 0.73$ | $55.14 \pm 0.82$ | $60.91 \pm 0.77$ | 55.19 | $54.26 \pm 0.73$ | $48.44 \pm 0.83$ | 51.18 | 53.51 |
| ER | 10.58 | $47.63 \pm 0.72$ | $52.92 \pm 0.84$ | $59.47 \pm 0.76$ | 52.90 | $51.12 \pm 0.79$ | $47.23 \pm 0.82$ | 49.10 | 51.31 |
| GEM | 8.62 | $31.59 \pm 0.75$ | $34.21 \pm 0.90$ | $43.19 \pm 0.90$ | 35.69 | $34.41 \pm 0.83$ | $31.94 \pm 0.77$ | 33.13 | 34.62 |
| LwF | 10.58 | $54.26 \pm 0.79$ | $58.43 \pm 0.77$ | $\mathbf{64.23 \pm 0.70}$ | $\mathbf{58.69}$ | $58.79 \pm 0.75$ | $\mathbf{51.26 \pm 0.77}$ | $\mathbf{54.77}$ | $\mathbf{57.06}$ |
| EWC | 8.77 | $50.13 \pm 0.79$ | $53.56 \pm 0.79$ | $60.18 \pm 0.75$ | 54.31 | $53.63 \pm 0.77$ | $47.05 \pm 0.79$ | 50.13 | 52.56 |
| RPSnet | $\mathbf{12.47}$ | $\mathbf{56.82 \pm 1.25}$ | $\mathbf{58.91 \pm 1.11}$ | $60.44 \pm 1.14$ | $\mathbf{58.69}$ | $\mathbf{62.75 \pm 1.11}$ | $44.53 \pm 1.22$ | 52.09 | 55.86 |
| MultiTask* | 79.34 | $69.19 \pm 0.68$ | $69.12 \pm 0.62$ | $70.05 \pm 0.63$ | 69.45 | $74.67 \pm 0.64$ | $52.88 \pm 0.78$ | 61.91 | 66.23 |
| Finetune* | 44.84 | $50.61 \pm 0.78$ | $54.66 \pm 0.78$ | $60.45 \pm 0.78$ | 54.95 | $54.17 \pm 0.77$ | $47.69 \pm 0.82$ | 50.72 | 53.18 |
| ER* | $\mathbf{65.43}$ | $50.58 \pm 0.74$ | $\mathbf{55.90 \pm 0.82}$ | $\mathbf{61.65 \pm 0.75}$ | $\mathbf{55.68}$ | $54.54 \pm 0.77$ | $\mathbf{48.98 \pm 0.82}$ | $\mathbf{51.61}$ | $\mathbf{53.98}$ |
| GEM* | 29.85 | $14.09 \pm 0.47$ | $16.67 \pm 0.48$ | $15.32 \pm 0.53$ | 15.29 | $15.31 \pm 0.49$ | $16.42 \pm 0.52$ | 15.85 | 15.51 |
| LwF* | 45.65 | $\mathbf{50.77 \pm 0.80}$ | $54.90 \pm 0.79$ | $60.91 \pm 0.73$ | 55.22 | $\mathbf{54.83 \pm 0.76}$ | $47.63 \pm 0.80$ | 50.98 | 53.44 |
| EWC* | 44.10 | $45.57 \pm 0.78$ | $47.70 \pm 0.78$ | $55.70 \pm 0.80$ | 49.29 | $48.25 \pm 0.80$ | $42.81 \pm 0.79$ | 45.37 | 47.64 |
| MNTDP* | 56.25 | $23.69 \pm 1.00$ | $21.97 \pm 0.87$ | $24.27 \pm 0.93$ | 23.27 | $26.31 \pm 0.94$ | $18.31 \pm 0.71$ | 21.59 | 22.57 |

## E.8 Performance on ViT Backbone

In this subsection, we investigate the compositional generalization capability for methods with ViT backbone. For MNTDP*, each module consists of one Multi-Head Attention (MHA) and one Feed-Forward Network (FFN) instead of one ResBlock. For RPSnet, each module consists of one FFN, and modules in the same layer share the same MHA. We find this structure has an advantage w.r.t. continual performance in our experiments. The results on CGQA are shown in Table 5 and the results on COBJ are shown in Table 6. Similar to that in the ResNet-18 backbone, LwF(*) and ER(*) perform well on both $H_n$ and $H_r$. RPSnet has good compositionality on COBJ, showing its superiority on real-world tasks. Moreover, we observe that in CGQA, $A_{sub} > A_{pro} > A_{sys}$, that, models generally have better accuracy on *sub* rather than *sys* and *pro*. This observation is opposite compared with ResNet-18 results shown in the main text (i.e., $A_{pro} > A_{sys} > A_{sub}$). This is because the selected attributes for concepts in *sub* are mainly texture attributes (e.g., color). The multi-head attention layer has a bias on shape attributes while the convolution layer has a bias on texture attributes [29], thus, models with ResNet-18 backbone are more sensitive to the texture difference, which is introduced in *sub*.

Table 6: Results on COBJ (10-way) with ViT backbone. Accuracy (%) $\pm$ 95% confidence intervals (%) over 300 tasks are reported for few-shot testing phases. Task-IL settings denote with a postfix "*".

| Methods | $A_{con}$ | $A_{sys}$ | $A_{pro}$ | $H_n$ | $A_{non}$ | $A_{noc}$ | $H_r$ | $H_a$ |
|---|---|---|---|---|---|---|---|---|
| MultiTask | 28.23 | 41.90 ± 0.63 | 36.58 ± 0.72 | 39.06 | 49.88 ± 0.82 | 35.69 ± 0.69 | 41.61 | 40.29 |
| Finetune | 13.93 | 37.26 ± 0.59 | 33.08 ± 0.71 | 35.05 | 41.76 ± 0.74 | 33.63 ± 0.67 | 37.25 | 36.12 |
| ER | 18.10 | 35.03 ± 0.63 | 30.76 ± 0.66 | 32.75 | 43.62 ± 0.78 | 33.25 ± 0.68 | 37.73 | 35.07 |
| GEM | 13.43 | 34.29 ± 0.59 | 31.56 ± 0.67 | 32.87 | 38.96 ± 0.68 | 31.50 ± 0.66 | 34.84 | 33.82 |
| LwF | 15.07 | 39.52 ± 0.61 | 35.18 ± 0.73 | 37.22 | 44.81 ± 0.80 | **35.23 ± 0.65** | 39.45 | 38.30 |
| EWC | 14.67 | 36.36 ± 0.62 | 32.12 ± 0.66 | 34.11 | 40.30 ± 0.70 | 33.17 ± 0.68 | 36.39 | 35.21 |
| RPSnet | **23.83** | **44.72 ± 1.05** | **41.37 ± 1.26** | **42.98** | **59.33 ± 1.42** | 33.49 ± 1.12 | **42.81** | **42.90** |
| MultiTask* | 46.40 | 41.78 ± 0.67 | 35.95 ± 0.72 | 38.64 | 53.19 ± 0.84 | 35.23 ± 0.68 | 42.39 | 40.43 |
| Finetune* | 41.97 | 37.53 ± 0.66 | 33.49 ± 0.71 | 35.39 | 41.45 ± 0.77 | 34.94 ± 0.67 | 37.92 | 36.61 |
| ER* | 41.10 | 38.23 ± 0.64 | 33.48 ± 0.69 | 35.70 | **44.95 ± 0.78** | 34.46 ± 0.63 | 39.01 | 37.28 |
| GEM* | 26.70 | 36.36 ± 0.62 | 33.00 ± 0.71 | 34.60 | 39.34 ± 0.69 | 32.18 ± 0.61 | 35.40 | 35.00 |
| LwF* | 43.90 | **40.30 ± 0.64** | **34.70 ± 0.75** | **37.29** | 44.83 ± 0.79 | **35.50 ± 0.71** | **39.62** | **38.42** |
| EWC* | 40.07 | 37.65 ± 0.62 | 33.22 ± 0.69 | 35.30 | 41.95 ± 0.72 | 33.72 ± 0.66 | 37.39 | 36.31 |
| MNTDP* | **45.50** | 22.74 ± 0.70 | 21.68 ± 0.75 | 22.20 | 31.52 ± 0.82 | 20.37 ± 0.63 | 24.75 | 23.41 |

Table 7: Results on CPIN (10-way) with ResNet-18 backbone. Accuracy (%) $\pm$ 95% confidence intervals (%) over 300 tasks are reported for few-shot testing phases. Task-IL settings denote with a postfix "*".

| Methods | $A_{con}$ | $A_{sys}$ | $A_{pro}$ | $H_n$ | $A_{non}$ | $A_{noc}$ | $H_r$ | $H_a$ |
|---|---|---|---|---|---|---|---|---|
| Finetune | 9.22 | 48.97 ± 0.88 | 42.06 ± 0.91 | 45.25 | 52.33 ± 0.96 | 15.56 ± 0.43 | 23.99 | 31.35 |
| ER | 28.82 | 59.16 ± 0.89 | 45.94 ± 1.05 | 51.72 | 60.11 ± 1.01 | 14.55 ± 0.42 | 23.43 | 32.25 |
| GEM | 9.19 | 51.13 ± 1.08 | 46.14 ± 1.04 | 48.51 | 55.16 ± 1.05 | 13.23 ± 0.38 | 21.34 | 29.64 |
| LwF | 9.21 | **64.45 ± 0.65** | **57.87 ± 0.67** | **60.98** | **65.73 ± 0.74** | 17.97 ± 0.51 | 28.22 | 38.59 |
| EWC | 9.37 | 53.92 ± 0.75 | 50.36 ± 0.73 | 52.08 | 56.64 ± 0.83 | 17.11 ± 0.46 | 26.28 | 34.93 |
| RPSnet | **45.88** | 55.33 ± 0.81 | 48.21 ± 0.63 | 51.53 | 58.21 ± 0.95 | **26.32 ± 0.33** | **36.25** | **42.56** |
| Finetune* | 75.37 | 61.57 ± 0.68 | **58.47 ± 0.66** | 59.98 | **64.30 ± 0.77** | 18.32 ± 0.45 | 28.52 | **38.65** |
| ER* | **86.46** | **62.22 ± 0.85** | 50.20 ± 0.94 | 55.57 | 63.23 ± 0.96 | 15.74 ± 0.42 | 25.21 | 34.68 |
| GEM* | 19.68 | 57.26 ± 0.71 | 49.65 ± 0.71 | 53.18 | 58.33 ± 0.75 | 16.30 ± 0.43 | 25.48 | 34.45 |
| LwF* | 73.24 | 59.99 ± 0.72 | 51.05 ± 0.75 | 55.16 | 60.44 ± 0.84 | 16.21 ± 0.42 | 25.56 | 34.94 |
| EWC* | 76.15 | 57.83 ± 0.65 | 53.11 ± 0.69 | 55.37 | 59.00 ± 0.77 | 18.01 ± 0.53 | 27.60 | 36.83 |

### E.9 CPIN Results

We also report the performance of CL algorithms on the CPIN benchmark in Table 7. Results are similar to those on the CGQA benchmark. The better performance on *non* comparing with *sys* and *pro* shows that the compositional generalization capability is hurt when the concepts are more fine-grained. The opposite relationship between *sys* and *pro* (in CGQA, $A_{pro} > A_{sys}$) shows that generalization becomes more difficult when increasing the number of combined concepts. All methods nearly fail on *noc* (less than 20% on 10-way, except RPSnet with 26.32%). This is because CPIN is more difficult than CGQA and reusable knowledge is rare from the continual learning phase and can hardly be reused for unseen concepts.

## F    Implemented Algorithms on Split-CIFAR100 Benchmark

In this section, we run our implemented algorithms (i.e., MultiTask(*), Finetune(*), ER(*), GEM(*), LwF(*), EWC(*), RPSnet, and MNTDP*) on Split-CIFAR100 benchmark and the results are reported in Table 8.

Table 8: Results on Split-CIFAR100 with the ResNet-18 backbone. Accuracy $(\%) \pm 95\%$ confidence intervals $(\%)$ over 8 independent runs are reported. Task-IL settings are denoted with a postfix "*".

| Method | Accuracy |
|---|---|
| MultiTask | $53.42 \pm 0.30$ |
| Finetune | $7.37 \pm 0.27$ |
| ER | $12.86 \pm 0.13$ |
| GEM | $13.43 \pm 0.21$ |
| LwF | $8.07 \pm 0.26$ |
| EWC | $7.41 \pm 0.35$ |
| RPSnet | $\mathbf{37.07 \pm 1.07}$ |
| MultiTask* | $79.87 \pm 0.25$ |
| Finetune* | $46.77 \pm 1.29$ |
| ER* | $61.23 \pm 0.54$ |
| GEM* | $60.14 \pm 0.41$ |
| LwF* | $\mathbf{70.41 \pm 0.74}$ |
| EWC* | $46.12 \pm 1.23$ |
| MNTDP* | $66.94 \pm 0.79$ |