# OpenReview forum: "Does Continual Learning Meet Compositionality? New Benchmarks and An Evaluation Framework"
_NeurIPS.cc/2023/Track/Datasets_and_Benchmarks — NeurIPS 2023 Datasets and Benchmarks Poster_

### Official Review · Reviewer_6pHG · 2023-07-04
**interesting problem and experiment set ups; but dataset is unrealistic**

**Rating:** 7
**Confidence:** 4
**Clarity:** 1. Understand that there is page limi…

**Strengths:**

1. The paper is generally well-written; but it still needs some minor improvements (see below).
2. The compositional generalization problem in continual learning is an important and interesting problem.
3. Multiple aspects of compositional generalization are covered and assessed.
4. The dataset design is well-structured and well-organized (some concerns are expressed below).

**Additional Feedback:**

see comments above

**Correctness:**

The datasets are limited to grid-like images for multi-task classification. These images are not naturalistic and may not reflect the nature of compositionality in an implicit manner. In other words, the compositionality in the datasets can be clearly defined as compositionality of class labels. The models might easily capture ``compositionality" by each grid cell. However, in naturalistic environment, the cues may not be explicit. This is the main concern of this paper. I would propose the following designs:

1. perform multi-task classification using naturalistic images, such as MSCOCO. For example, the task is to classify cars, pedestrian, and buses at the same time.
2. Cut and paste multiple classes of objects in context-relevant backgrounds
3. Introduce noise/background patches or distracting images from other classes to fill in the blank grid cells. The blank cells provide useful and implicit signals for models to focus on the remaining cells containing target class labels, which makes the problem unrealistic and trivial to solve. This is also demonstrated by GRADCAM; as the models are always paying attention to object regions.
4. I wonder whether the authors perform augmentations on these grid-like images during training; such as changing grid locations and expanding or shrinking grid sizes from 2x2 to 5 by 5 for example with some distractor image patches.

**Documentation:**

I checked the availability of the code and the dataset; however, there are no detailed instructions on how to use the code and datasets.

**Ethics:**

No ethical concerns, as far as I can see.

**Limitations:**

See the limitations and suggestions below.

**Opportunities For Improvement:**

See the limitations and suggestions below.

**Relation To Prior Work:**

There are several continual learning methods with augmented memory missing; such as [1] [2] and [3]. See [4] for a full review of continual learning methods. Please include [1-3] as baselines in the benchmark.

[1] Lifelong Compositional Feature Replays Beat Image Replays in Stream Learning
[2] Variational prototype replays for continual learning
[3] REMIND Your Neural Network to Prevent Catastrophic Forgetting
[4] Continual lifelong learning with neural networks: A review

**Summary And Contributions:**

The paper aims to address the problem of benchmarking compositional generalization in continual learning settings. Two benchmark datasets and one evaluation framework are proposed. Three aspects of compositional generalization are proposed: systematicity, producibility, and substitutivity. The results on seven continual learning methods are tested.

---

> ### Author Response · Authors · 2023-08-21
> **To Reviewer 6pHG (1/3)**
>
> ### Q1: The datasets are limited to grid-like images for multi-task classification and images are not naturalistic
> >
> > We would like to highlight the key difference between our CFST and the multi-task classification / multi-label recognition task, which we have already illustrated in Figure 1 and Remark 3.2 [Lines 106-111] in the main text.
> >
> > - In CFST **only the image-level label is available and no concept label is provided**, which is much more challenging than multi-label classification where labels of all concepts in an image are accessible.
> >
> > - Thus, CFST requires **inference of the underlying compositional concepts behind an image**, while multi-label classification does not encourage compositional learning as we have detailed in Appendix A.
> >
> >   - Capturing those discriminative features only for classification (e.g., the shape that differentiates "horse" from "human") and failing to capture all the compositional features (e.g., the fine-grained features helpful for identification of "horse") likely struggles in a future CFST task (e.g., classifying between "zebra" and "horse").
> >
> > The objective of our benchmark is exactly to evaluate the capability of a model in compositional learning, i.e., **whether it understands the previously learned concepts and generalizes to future tasks**. We underline that CGQA and COBJ meet this objective in practice:
> >
> > - A combined image with only a single label in CGQA is **the same as a natural image in practical applications**, except that we aim to construct a dataset like CGQA with **(1)** all the concepts in an image **easy to parse and interpret**, and **(2)** **a comparably small number of concepts**. CGQA constructed in such a manner serves as a **less challenging benchmark** to evaluate various methods.
> >
> > - The **more challenging benchmark of COBJ** contains images that also have **single labels without concept/object labels**; compared to CGQA, the concepts within each image are more complicated and realistic.
> >
> >   - We have detailed the construction process of COBJ in Appendix C that we crop the region with target concepts on one image, thus, images in COBJ are not grid-like;
> >
> >   - We have provided image examples in Appendix Figure 5.
> >
> > By the way, thank you very much for your following designs about other methods to construct benchmarks, which really inspire me and we will carefully consider them.

---

> > ### Author Response · Authors · 2023-08-21
> > **To Reviewer 6pHG (2/3)**
> >
> > ### Q2: About augmentations during training
> > >
> > > - In our experiments, we perform the random horizontal flipping for methods with ResNet-18 backbones and additionally perform the Rand-Augment and random erasing for methods with ViT backbones, which we have already summarized in Appendix D lines 425-434.
> > >
> > > - We also try to perform **augmentation of randomly permutating grid locations**. The results are shown below, and for comparison, we also list the results we reported in the main text:
> > >
> > >   | Aug       |   continual | sys           | pro           | sub           |    Hn | non           | noc           |    Hr |    Ha |
> > >   |:----------|------------:|:--------------|:--------------|:--------------|------:|:--------------|:--------------|------:|------:|
> > >   | Finetune  |        8.42 | 63.96 +- 1.73 | 67.22 +- 1.75 | 59.36 +- 1.85 | 65.55 | 67.58 +- 2.13 | 39.34 +- 2.04 | 49.73 | 56.55 |
> > >   | ER        |       18.05 | 73.86 +- 1.63 | 75.18 +- 1.62 | 63.86 +- 1.81 | 74.51 | 77.72 +- 1.54 | 40.92 +- 1.85 | 53.61 | 62.36 |
> > >   | Finetune* |       72.13 | 67.66 +- 2.07 | 71.28 +- 1.62 | 62.06 +- 1.74 | 69.42 | 72.32 +- 1.78 | 40.22 +- 1.84 | 51.69 | 59.26 |
> > >   | ER*       |       77.88 | 72.28 +- 1.66 | 75.42 +- 1.47 | 67.82 +- 1.52 | 73.82 | 77.50 +- 1.72 | 43.62 +- 1.81 | 55.82 | 63.57 |
> > >
> > >   | Paper     |   continual | sys           | pro           | sub           |    Hn | non           | noc           |    Hr |    Ha |
> > >   |:----------|------------:|:--------------|:--------------|:--------------|------:|:--------------|:--------------|------:|------:|
> > >   | Finetune  |        8.38 | 64.73 +- 0.78 | 65.43 +- 0.73 | 61.26 +- 0.67 | 63.75 | 68.54 +- 0.80 | 40.32 +- 0.72 | 50.77 | 57.84 |
> > >   | ER        |       19.78 | 71.38 +- 0.75 | 70.11 +- 0.64 | 64.32 +- 0.69 | 68.46 | 77.27 +- 0.67 | 40.98 +- 0.72 | 53.56 | 61.60 |
> > >   | Finetune* |       72.46 | 70.32 +- 0.73 | 72.62 +- 0.63 | 66.33 +- 0.69 | 69.66 | 75.32 +- 0.70 | 43.26 +- 0.73 | 54.95 | 62.92 |
> > >   | ER*       |       76.05 | 71.37 +- 0.70 | 72.67 +- 0.69 | 66.80 +- 0.63 | 70.19 | 76.28 +- 0.66 | 45.61 +- 0.77 | 57.09 | 64.29 |
> > >
> > >   - The results are quite similar with and without changing grid locations.
> > >
> > > - As for **expanding grid sizes and introducing some distractor image patches**, we do not recommend doing that. The reasons are as follows:
> > >
> > >   - The provided image-level labels are the existence of all concepts in the images, thus, introducing other image patches may potentially change the number of existing concepts and change the label.
> > >
> > >   - While our Productivity test aims to evaluate the generalization capability on more concepts, we recommend preventing the model from potentially seeing more concepts in one image during continual training.
> > >
> > > - By the way, we also tried another benchmark with the 3x3 grid-like images, called CPIN, constructed from PartImageNet. The details and results have been presented in Appendix C.1.7 and E.9, respectively. Since it provides similar conclusions as CGQA, so we only present CGQA in the main paper.

---

> > > ### Author Response · Authors · 2023-08-21
> > > **To Reviewer 6pHG (3/3)**
> > >
> > > ### Q3: About the construction process
> > > >
> > > > - Thank you very much for your understanding of the page limit. We have provided a short description in Sec 5 [lines 193-198] about the source dataset we used.
> > > >
> > > > - We also clearly understand that the detailed construction processes are important for readers to understand our work. We will put these parts to the main text in our revision.
> > >
> > > ### Q4: Missing literature review about augmented-memory-based continual learning
> > > >
> > > > - Thank you very much for providing me with these papers. I will include them and improve the related works part. We will submit our revision as soon as possible.
> > > >
> > > > - Additionally, we run REMIND on our 10-way COBJ, and the results are shown below:
> > > >
> > > >   | COBJ      | Acon| sys | pro | Hn | non | noc | Hr | Ha |
> > > >   | --------- | --- | --- | --- | --- | --- | --- | --- | --- |
> > > >   | REMIND    | 25.70 | 34.20 ± 1.82 | 30.56 ± 1.74 | 32.28 | 40.64 ± 2.21 | 29.22 ± 1.41 | 34.00 | 33.11 |
> > > >   | Finetune  | 17.60 | 37.56 ± 0.86 | 29.98 ± 0.88 | 33.34 | 47.00 ± 1.03 | 28.41 ± 0.84 | 35.41 | 34.34 |
> > > >   | LwF       | 18.40 | 45.50 ± 0.87 | 38.16 ± 0.98 | 41.50 | 51.84 ± 0.95 | 34.08 ± 0.87 | 41.12 | 41.31 |
> > > >
> > > >   - REMIND has better **Acon** than other baselines (of course, we have not carefully tuned the hyper-parameters such as lr and the number of replay samples. We believe its performance can be further improved). And compositionality (i.e., **Hn**) does not outperform others.
> > > >
> > >
> > > Thank you again for your comments.

---

> > > > ### Comment · Reviewer_6pHG · 2023-08-25
> > > > **Acknowledgement of reading the authors' responses**
> > > >
> > > > I would like to thank the authors for clarifying my doubts. The responses provided by the authors adequately addressed my concerns. This is an indeed interesting problem setting and a reasonable setup, which can be potentially used in real-world applications.
> > > >
> > > > Initially, I got confused by Fig1 (I believe that reviewer xdij also raised a similar confusion and I got to think this way after reading all the other reviewers' comments). I suggest that you remove the 0 and 1 for all the labels and simplify figure designs to make them align better with the proposed problem settings.
> > > >
> > > > Another comment that I initially forgot to ask the authors:
> > > > In addition to forgetting, BWT [2] nd FWT [1] are also useful and essential metrics for continual learning. It would be interesting to include results with these two standard evaluation metrics in the final version. These results can tell us how much positive and backward compositional knowledge transfer happens across tasks.
> > > >
> > > > [1] Sen Lin, Li Yang, Deliang Fan, and Junshan Zhang. Beyond not-forgetting: Continual learning with backward knowledge transfer. Advances in Neural Information Processing Systems, 35:16165–16177, 2022.
> > > >
> > > > [2] David Lopez-Paz and Marc’Aurelio Ranzato. Gradient episodic memory for continual learning. Advances in neural information processing systems, 30, 2017
> > > >
> > > > I would like to raise my scores to 7.

---

### Official Review · Reviewer_fvLh · 2023-07-17
**A new and comprehensive benchmark for continual compositionality learning**

**Rating:** 5
**Confidence:** 4

**Strengths:**

1. The evaluation of model's compositional generalizability is useful for the research community.
2. The proposed evaluation procedure is comprehensive with 5 different protocols to assess model's compositionality from different angles.
3. The finding that modularity-based methods lack compositional generalization is interesting.

**Additional Feedback:**

- Minor grammatical mistakes such as "presents" on Line 102.
- Provide a definition of a label space $Y_{tr}$ and $Y_{nv}$.
- Explain why we need to adopt attribute factorization to model data generation. Improve the flow of Section 4.

**Clarity:**

I do not see the purpose of a section about "concept factorization". More clear explanations why we need to model the data generation using attribute factorization should be provided. Section 4 is fragmented and lack a coherent flow.

Other sections are clear and written well.

**Correctness:**

The claim about "compositionality addresses the stability-plasticity dilemma" is not substantiated via the experiments. An experimental result in Table 1 contradicts to the claim.

**Documentation:**

Yes, URL to dataset is included.

**Ethics:**

No ethics concern.

**Limitations:**

1. The main motivation of the paper, as indicated in the Introduction, is "compositionality addresses the stability-plasticity dilemma" (Line 17-18). However, the experiments to support this claim are insufficient. First, from Table 1, LwF has the best compositional ability, but low continual learning accuracy. It contradicts with the aforementioned motivation of this paper. From the analysis, I speculate that compositionality and continual learning ability are the two independent aspects of machine learning. Then, why do need to measure the compositionality if it doesn't improve the continual learning ability of the model?
2. To verify the claim in the Abstract "compositionality helps continual learning", the paper should also conduct a truly continual learning setting, i.e., none of the previous concepts are present in the next task. Current continual learning setting randomly groups a pairs of "concepts", which results in overlapping concepts between tasks, i.e., leading to knowledge leaking. However, the standard continual learning setting only evaluates on novel classes, where no overlapping and knowledge leaking are present
3. The evaluations of 3 compositional capabilities lack motivations. Why are we interested in model's systematicity, productivity and substitutivity?
4. The adoption of few-shot learning and freezing feature extractor in evaluating compositionality also lacks motivations. How does few-shot learning (Principle 2 in the introduction) and frozen feature extractor (Principle 3) support compositional testing.
5. The literature review misses several recent works. For example, NeurIPS2022 [1] and CVPR2023 [4] works also proposes a benchmark for testing mode's compositionality. Additionally, the benchmark should include more recent continual learning works [2,3] to give a more meaningful conclusion about the current model's compositionality. Especially, a recent method [4] also explores compositionality in continual learning, which should be included in the benchmark comparisons and the literature review.
6. Some experimental findings are unclear. Why is forgetting is not as suffered as that in the class-IL setting on CGQA (Line 244-245)?
7. The explanation of high $A_{sub}$ on ViT and low $A_{sub}$ on conv-based methods is unclear. Conv-based methods are more sensitive to texture, and selected attributes for `sub` are mainly texture attributes. Hence, should conv-based methods be better than Transformer for $A_{sub}$?
8. From Fig. 3, why does $S(sys) have positive values when using around 0 to 100 samples? My understanding is that A_non is always better than A_sys as A_non is tested only on seen concept combination. Could you please explain why S(sys) are positive when few samples (between 0 to 100 samples) are present?
9. From Fig. 4, it can be seen that having smaller number of classes with more tasks reduce the accuracy. It contradicts with the observation on Line 320-321: "the learner needs a smaller number of compositional features necessary for distinguishing classes in each task"

References:
[1] Zerroug et al., "A Benchmark for Compositional Visual Reasoning", NeurIPS, 2022.

[2] Wang, Zifeng, et al., "Dualprompt: Complementary prompting for rehearsal-free continual learning", ECCV, 2022.

[3] Wang et al., "S-prompts learning with pre-trained transformers: An occam’s razor for domain incremental learning", NeurIPS, 2022

[4] Smith, James Seale, et al., "Construct-vl: Data-free continual structured vl concepts learning", CVPR, 2023.

**Opportunities For Improvement:**

1. Justify the claim in the Introduction about "compositionality addresses the stability-plasticity dilemma" (Line 17-18) via experimental results on unseen class continual learning. Explain the contradiction between the findings in Table 1 and the paper's claim (Line 17-18).
2. Articulate the motivations on the evaluations of 3 compositional capabilities.
3. Explain why Principle 2 (few-shot learning) and Principle 3 (frozen feature extractor) supports evaluating model's compositionality.
4. Provide literature review and experiment results of more recent works in continual learning (see Point 5 in the Limitations section).
5. Provide more justification on the claim "forgetting is not as suffered as that in the class-IL setting on CGQA (Line 244-245)".
6. Explain why conv-based methods have lower $A_{sub}$ even though they are sensitive to texture information, and sub protocol also uses texture information to composite images.
7. See suggestions in Point 8 in Limitations section.
8. Explain the contradictions as indicated in Point 9 in Limitations section.
9. Explain the overview of CFST evaluation protocol in Section 4, what the flow is before describing the details.

**Relation To Prior Work:**

The literature review misses several recent works. It should compare with NeurIPS2022 [1] and CVPR2023 [4] works, which also introducing a benchmark for testing model's compositionality.

The benchmark should include more recent continual learning works [2,3] to give a more meaningful conclusion about the current model's compositionality. Especially, a recent method [4] also explores compositionality in continual learning, which should be included in the benchmark comparisons and the literature review.

References:
[1] Zerroug et al., "A Benchmark for Compositional Visual Reasoning", NeurIPS, 2022.

[2] Wang, Zifeng, et al., "Dualprompt: Complementary prompting for rehearsal-free continual learning", ECCV, 2022.

[3] Wang et al., "S-prompts learning with pre-trained transformers: An occam’s razor for domain incremental learning", NeurIPS, 2022

[4] Smith, James Seale, et al., "Construct-vl: Data-free continual structured vl concepts learning", CVPR, 2023.

**Summary And Contributions:**

The paper presents an benchmark to evaluate model's compositionality. The dataset composites up to 4 images to generate a combination of concepts. The paper uses five composition procotols to test 3 aspects of compositionality, i.e., systematicity, productivity, substitutivity and evaluate two reference performance, i.e., non-novel testing (the upper-bound) and unseen concept testing (the lower-bound).It conducts experiments on five baselines and two modular-based methods a using few-shot learning paradigm.

---

> ### Author Response · Authors · 2023-08-21
> **To Reviewer fvLh (1/5)**
>
> We sincerely appreciate your constructive comments on this paper. We detail our response below point by point. Please kindly let us know if our response addresses the issues you raised in this paper.
>
> ### Q1: Contradiction between experimental results and the claim: "Compositionality addresses the stability-plasticity dilemma"
> >
> > First, we would like to highlight that we are discussing the stability and plasticity of the **feature extractor** of a continual learner. In this case, **average test accuracy is not a suitable metric since it takes account of both the feature extractor and the classifier**.
> > - We analyze the forgetting phenomenon of the average test accuracy as follows:
> >   1. The **feature extractor** forgets the crucial features for old tasks when learning the new task.
> >   2. In a multi-classifier implementation: the **classifiers** of old tasks can not update (no old sample); in a single-classifier implementation: the prediction will bias to classes in the current task (non-i.i.d). Thus, the coupling between the feature extractor and the corresponding classifiers is broken.
> > - In our experiment, we visualized the CAM of the feature extractor in Appendix E.3. On CGQA, the learned feature extractor of Finetune was compositional, thus, the feature extractor had good stability. However, Acon was very bad, showing that **when combined with the classifiers, the stability was poor**. Thus, we claim that using Acon to evaluate the feature extractor is faulty.
> > - While our Hn evaluates the compositionality of a feature extractor (how well can this feature extractor extract compositional features). We eliminate the effect of the classifier.
> > - By the way, our evaluation method is flexible and can also be used on algorithms that don't explicitly separate feature learning from classifier learning. We just provide few-shot testing tasks and algorithms can just deepcopy and evaluate their models with their own methods.
>
> ### Q2: Motivations on the evaluations of three compositional capabilities
> >
> > - First, we would like to humbly clarify that compositionality is a very important ability for a continual learner. And most of the current works (see Sec 2 related works) in vision **only consider systematicity (novel re-combination)** as compositionality. We extend to productivity and substitutivity (other two very interesting aspects of compositionality which are widely studied in the NLP field) to provide more insights.
> >
> > - We now discuss more about the motivations for these three compositional capabilities:
> >
> >   - **Systematicity**: This is the most general aspect of compositionality which question models whether they can understand novel-recombination of seen concepts. For example, an **un-compositional** feature extractor may learn **coupled features** between Grass and Table concepts when recognizing {Grass, Table}. Then, when seeing {Shirt, Table} (one image with the Table concept but no Grass concept), it does not have high activating values on the corresponding features. On the other hand, a **compositional** feature extractor learns **decoupled features for Grass and for Table concepts separately**. Thus, it can have higher activating values on Table features when seeing {Shirt, Table}.
> >
> >   - **Productivity**: If a feature extractor has good compositionality, the extracted features exactly represent each compositional component. Then **it should be easy to generalize to complex images with more seen concepts**. For example in our main paper lines 150-151, after gathering knowledge of concepts Door, Shirt, Grass, Table, Hat, Leaves from the task (distinguishing {Door, Shirt}, {Grass, Table}, and {Hat, Leaves}), the model should easily understand {Door, Leaves, Shirt, Table}, although it does not seen any instance of this label before. Our productivity test tasks only contain images with more concepts than the continual training tasks (e.g., images in continual training tasks may only have 2 concepts while images in the productivity test tasks have 3 or 4 concepts).
> >
> >   - **Substitutivity**: In order to achieve **balance** and **flexible** combinations of concepts (the number of instances for different combinations of concepts can be similar (no long-tailed combinations) and we can combine any pair of concepts), our selected concepts are all visual and disentangled. However, some concepts (e.g., white color) are more likely to be the ''attribute'' of other concepts (e.g., shirt). **These attribute-like concepts are not so flexible and they sometimes accompany by some other concrete concepts**. To compensate for the evaluation of these attribute-like concepts, we design the substitutivity test.

---

> > ### Author Response · Authors · 2023-08-21
> > **To Reviewer fvLh (2/5)**
> >
> > ### Q3: The proposed benchmarks are not truly in the continual learning setting and knowledge leaking on continual training tasks
> > >
> > > We are very sorry that our description makes you think we did not conduct a truly continual learning setting. We would like to humbly clarify that **our setting is also a standard CL setting**, which we have illustrated in Figure 1 and Remark 3.2 [Lines 106-111] in the main text.
> > >
> > > - In CFST **only the image-level label is available and no concept label is provided**, thus, CFST requires **inference of the underlying compositional concepts behind an image**.
> > >
> > > - We compare our benchmarks with a standard CL setting: Split-CIFAR100. Our CGQA has 100 different labels and if the number of tasks is 10, each task will have 10 different labels. These labels are actually the existence of the concepts, thus, different labels may contain overlapped concepts (e.g., there can be two labels: {Door, Shirt} and {Grass, Shirt}). As pointed out in Sec 3 Remark 3.2, lines 106-111, and Figure 1, these concepts are potentially hidden. For example, we can assign label 0 to {Door, Shirt} and label 1 to {Grass, Shirt}. They are totally different labels and we do not tell the models that label 0 and label 1 all have the Shirt concept.
> > >
> > >   - This is just the same as Split-CIFAR100 with 100 labels and these labels are evenly distributed in 10 tasks.
> > >
> > >   - There are two labels (i.e., pine_tree, oak_tree) in CIFAR100. They can be assigned to different tasks in the Split-CIFAR100 setting. These two labels also have overlapped concepts (e.g., leaves, trunk), and the concepts are potentially hidden.
> > >
> > > - By the way, we also evaluate unseen concepts on noc (non-compositional testing) as a reference.
> >
> > ### Q4: Explain why Principle 2 (few-shot learning) and Principle 3 (frozen feature extractor) supports evaluating the model's compositionality.
> > >
> > > In order to evaluate the model's compositionality, we exactly try to question the model **whether it understands the previously learned concepts and generalizes to future tasks (unseen re-combination of concepts)**, which we have already listed the reasons in the main paper line 120-126.
> > >
> > > - At the specific checkpoint (e.g., after finishing continual training tasks), we adopt additional evaluation tasks for this purpose. In such a condition, if the number of support samples in the evaluation task is large, the feature extractor may learn from them and thus we can not actually judge **whether the good performance comes from the original feature extractor (learned from old tasks)**. Thus, we recommend few-shot evaluation tasks and frozen feature extractors.
> > >
> > > - We should highlight that **the principles are not strict** and we also did experiments on not frozen feature extractors in Appendix E.7. For your convenience, we show our observation: all methods show a performance drop if not freezing the feature extractor, especially for ER. It is clearly an overfitting issue and the bad effect is method-dependent. So in order to eliminate this effect when comparing the methods and let the accuracy correctly represents the compositionality, we freeze the feature extractor.
> > >
> > > - On the other hand, **few-shot tasks can help evaluate the plasticity** since models can have good accuracy on these tasks only if they can fast adapt previous knowledge to the new one. In this case, the compositionality is crucial to fill the systematic gap between the continual training and the few-shot testing tasks.

---

> > > ### Author Response · Authors · 2023-08-21
> > > **To Reviewer fvLh (3/5)**
> > >
> > > ### Q5: Add literature review of more recent works and baseline experiments
> > > >
> > > > - Thank you very much for providing recent works. I will put them into our related works. We will submit our revision as soon as possible.
> > > > - We also run quick experiments on codaPrompt, dualPrompt, l2p++, deep l2p++, and the corresponding finetune method with pretrained backbone (**FT_Classifier**: freeze feature extractor and finetune classifier; **l2p++**: use prefix-tuning instead of prompt-tuning; **deep l2p++**: add prefix-tuning at all layers). The results are as follows:
> > > >
> > > >   | CGQA          | Acon| sys | pro | sub | Hn | non | noc | Hr | Ha |
> > > >    |---------------| --- | --- | --- | --- | --- | --- | --- | --- | --- |
> > > >    | dual-prompt   | 85.52 ± 1.47 | 65.98 ± 1.68 | 69.32 ± 1.61 | 76.72 ± 1.56 | 70.40 | 69.26 ± 1.63 | 84.56 ± 1.22 | 76.15 | 72.59 |
> > > >    | coda-prompt   | 77.43 ± 1.91 | 52.24 ± 1.46 | 53.96 ± 1.77 | 62.14 ± 1.60 | 55.80 | 54.50 ± 1.56 | 74.38 ± 1.76 | 62.91 | 58.44 |
> > > >    | l2p++         | 83.02 ± 1.66 | 61.46 ± 1.54 | 63.02 ± 1.61 | 71.28 ± 1.60 | 64.98 | 64.72 ± 1.75 | 81.70 ± 1.42 | 72.23 | 67.70 |
> > > >    | deep l2p++    | 77.84 ± 1.91 | 52.22 ± 1.60 | 54.52 ± 2.00 | 62.70 ± 1.71 | 56.14 | 54.24 ± 1.58 | 74.24 ± 1.50 | 62.68 | 58.58 |
> > > >    | FT_Classifier | 78.13 ± 1.85 | 52.34 ± 1.51 | 54.04 ± 1.28 | 61.04 ± 1.47 | 55.56 | 54.14 ± 1.69 | 74.40 ± 1.82 | 62.67 | 58.20 |
> > > >
> > > >   | COBJ          | continual | sys | pro | Hn | non | noc | Hr | Ha |
> > > >    |---------------| --- | --- | --- | --- | --- | --- | --- | --- |
> > > >    | dual-prompt   | 90.00 ± 3.63 | 62.48 ± 1.97 | 48.94 ± 2.62 | 54.89 | 51.80 ± 2.36 | 84.04 ± 1.46 | 64.09 | 59.13 |
> > > >    | coda-prompt   | 89.20 ± 3.89 | 60.46 ± 2.10 | 46.76 ± 2.68 | 52.73 | 49.84 ± 2.24 | 83.28 ± 1.61 | 62.36 | 57.14 |
> > > >    | l2p++         | 89.37 ± 3.46 | 61.52 ± 2.11 | 47.50 ± 2.59 | 53.61 | 50.60 ± 2.49 | 83.48 ± 1.42 | 63.01 | 57.93 |
> > > >    | deep l2p++    | 89.90 ± 3.35 | 60.68 ± 2.04 | 46.54 ± 2.55 | 52.68 | 49.22 ± 2.33 | 82.78 ± 1.38 | 61.73 | 56.85 |
> > > >    | FT_Classifier | 89.07 ± 3.98 | 60.62 ± 1.97 | 46.78 ± 2.69 | 52.81 | 48.96 ± 2.25 | 83.44 ± 1.36 | 61.71 | 56.91 |
> > > >
> > > >  - These prompt-based methods utilize a pre-trained backbone and learn to extract knowledge from the backbone by prompting. It is clear that these pre-trained methods have better **Acon** and better **Hr** than the methods we reported in the main text. However, **Hn** does not outperform and even is worse than the methods we reported in the main text, which indicates that **the good test accuracy comes from the strong pre-trained backbone, however, the compositionality is not better**.
> > > >
> > > >  - Further, the good **noc** shows that they have potentially seen these concepts before (those from-scratch learning methods reported in our paper generally have poor **noc** performance).
> > > >
> > > >    - Thus, we claim that **the pretrained backbone may potentially see the labels for testing before which is unfair** to those from-scratch learning methods (baselines I used in our experiments).
> > >
> > > ### Q7: Explain why conv-based methods have lower (A_sub) even though they are sensitive to texture information, and sub protocol also uses texture information to composite images.
> > > >
> > > > - ''Conv-based models are sensitive to texture information'', thus, they **tend to use texture features for prediction**. Further, when texture features are absent from the target concept (in the sub test, we use objects with different texture features for evaluation, e.g., train using red, black, and white shirts but test the green shirt in sub), models are confused to recognize the target concept.
> > > >   - For example, a model recognizes the shirt concept by its color “red or black or white”. When a “green shirt” comes, this model does not recognize that it is also a shirt. Thus, it results in poor test acc on the sub test.
> > > > - On the contrary, **vit-based models tend to use shape features for prediction** and can correctly recognize “green shirt” as the shirt concept since it has the same shape as other shirts.
> > > > - Note that we guaranteed the Solvability that the evaluated attributes are seen in other concepts. The poor $A_{sub}$ results indicate its pool compositionality on the attribute level.

---

> > > > ### Author Response · Authors · 2023-08-21
> > > > **To Reviewer fvLh (4/5)**
> > > >
> > > > ### Q6: Provide more justification on the claim "forgetting is not as suffered as that in the class-IL setting on CGQA (Line 244-245)".
> > > > >
> > > > > - Sorry, my wrong grammar leads to the misunderstanding. We would like to correct our claim: ''**This is because forgetting on CGQA is not as suffered as that on COBJ, especially in the task-IL setting.**'' We will update this in our revision.
> > > > >
> > > > > - To justify this, we show the test accuracies for Finetune **just after finishing each continual training task** as follows:
> > > > >   - task-IL 10-way CGQA tasks
> > > > >
> > > > >     | evaluate on task | 1 | 2 | 3 | 4 | 5 | 6 | 7 | 8 | 9 | 10 |
> > > > >     |---------------| --- | --- | --- | --- | --- | --- | --- | --- | --- | --- |
> > > > >     | finish task 1 | 58.4 | 0 | 0 | 0 | 0 | 0 | 0 | 0 | 0 | 0 |
> > > > >     | 2             |55.2 | 66.7 | 0 | 0 | 0 | 0 | 0 | 0 | 0 | 0 |
> > > > >     | 3             | 55.1 | 65.4 | 74.4 | 0 | 0 | 0 | 0 | 0 | 0 | 0 |
> > > > >     | 4             | 54.2 | 63.3 | 61.3 | 83.8 | 0 | 0 | 0 | 0 | 0 | 0 |
> > > > >     | 5             | 54.7 | 56.4 | 59.4 | 71.9 | 75.4 | 0 | 0 | 0 | 0 | 0 |
> > > > >     | 6             | 48.9 | 57.7 | 66.0 | 71.1 | 72.2 | 77.1 | 0 | 0 | 0 | 0 |
> > > > >     | 7             | 54.0 | 58.7 | 63.8 | 75.9 | 66.6 | 72.4 | 75.3 | 0 | 0 | 0 |
> > > > >     | 8             | 47.2 | 58.4 | 54.1 | 74.3 | 64.9 | 71.1 | 71.0 | 74.8 | 0 | 0 |
> > > > >     | 9             | 61.6 | 70.0 | 68.2 | 82.9 | 76.4 | 76.4 | 75.5 | 69.9 | 84.8 | 0 |
> > > > >     | 10            | 53.4 | 69.1 | 74.3 | 78.3 | 75.5 | 75.5 | 70.1 | 67.1 | 82.5 | 82.2 |
> > > > >
> > > > >   - task-IL 10-way COBJ tasks
> > > > >
> > > > >     | evaluate on task | 1 | 2 | 3 |
> > > > >     |---------------| --- | --- | --- |
> > > > >     | finish task 1 | 54.1 | 0 | 0 |
> > > > >     | 2             |28.1 | 55.1 | 0 |
> > > > >     | 3             | 35.5 | 29.5 | 53.3 |
> > > > >
> > > > > - On CGQA, forgetting is relatively smaller than COBJ even for the naive Finetune method. It is intuitive since COBJ is a real-world benchmark and CGQA is a synthesized benchmark.
> > > > > - The key point of MNTDP* to address catastrophic forgetting is to freeze old modules, thus, **it can achieve no forgetting**. However, **in our CGQA case, forgetting is not as suffered as the real-world benchmark (e.g., COBJ)**. Thus, the advantages of MNTDP* are not obvious on CGQA. However, **in COBJ, MNTDP can largely eliminate forgetting**, thus, outperforming the others.
> > > >
> > > > ### Q8: Experimental results on ''Sample efficiency for learning compositionality'': why S(sys) is positive when few samples (between 0 to 100 samples) are present?
> > > > >
> > > > > - First, I should clarify that the samples here refer to **training samples in the continual training tasks** (lines 306-307).
> > > > >
> > > > > - Note that $S(sys)=(A_{sys}-A_{non})/A_{non}$. Here non-novel (non) testing tasks contain the same number of labels as training tasks, but the K labels are randomly chosen from the training label pool and there is no intersection between their sample instances.
> > > > >
> > > > >   - Thus, when **the model is not well-trained (which is the case when the number of training samples for each continual task is very few (less than 100))**, $S(sys)$ can be very unstable and $A_{non}$ does not necessarily better than $A_{sys}$.

---

> > > > > ### Author Response · Authors · 2023-08-21
> > > > > **To Reviewer fvLh (5/5)**
> > > > >
> > > > > ### Q9: experimental results on ''Varying number of continual training tasks'': why the small-way task needs a smaller number of compositional features for distinguishing classes but the accuracy drops when decreasing the number of classes in the task.
> > > > > >
> > > > > > First, we would like to clarify that **we vary the number of classes in the continual training tasks** and **the reported Hn is the result for models on the 10-way compositional testing tasks** since we need to keep the same number of classes for comparison.
> > > > > >
> > > > > > - For a specific continual task, the model will learn crucial compositional features but miss other compositional features which are not needed for this task but may be crucial for future tasks.
> > > > > >
> > > > > >   - Taking the example in Appendix E.5 lines 613-616, one can distinguish a horse from a person by their different shapes. But this is not enough for the case of horse and zebra (i.e., limited compositionality). However, for the tri-classification task of distinguishing between horse, zebra, and person, one can learn both shape and texture features (i.e., relatively better compositionality). The learned texture features can be used in future tasks like distinguishing tigers from other animals.
> > > > > >
> > > > > > - Thus, the model may not obtain the necessary features for these compositional testing tasks during the continual training phase. As a result, the performance of evaluating compositionality (i.e., Hn) drops.
> > > > >
> > > > > ### Q10: Purpose to use “concept factorization”.
> > > > > >
> > > > > > Specifically, we need to mathematically describe the data generation process from the source datasets (i.e., GQA and Objects365).
> > > > > >
> > > > > > - The key differences between training tasks and testing tasks are the marginal concept distributions and the conditional distributions which can be clearly modeled with ''concept factorization''.
> > > > > >
> > > > > >   - **Marginal distributions**: Different combinations of concepts form the different distributions of the label spaces and are presented in different task sets.
> > > > > >
> > > > > >   - **Conditional distributions**: Substitutivity test uses $p_{sub}(x|c^{1:M})$ instead of $p(x|c^{1:M})$ since we test concepts with different attributes.
> > > > >
> > > > > Thank you again for your comments.

---

> > > ### Comment · Reviewer_fvLh · 2023-08-21
> > >
> > > Thanks for the clarification. I'm satisfied with the response.
> > >
> > > Please include the clarification of label setting in Q3. The current version can mislead people in the continual learning community.

---

> > > > ### Author Response · Authors · 2023-08-26
> > > > **Thank you for your reply**
> > > >
> > > > We sincerely appreciate your reply for acknowledging our motivations and pointing out the remaining concern. Please find our response below, and kindly let us know if it satisfactorily addresses your remaining concern.
> > > > ​
> > > > > - First and foremost, we would like to humbly clarify that we are not the first to draw the conclusion that ''compositionality addresses the stability-plasticity dilemma''; it **has been proposed or observed** in a couple of continual learning literatures including those for NLP [1,2,3] and for CV [4,5,6].
> > > > >
> > > > > - Secondly, compared to the abovementioned literature, **we are the first** to
> > > > >
> > > > >   - propose a novel evaluation protocol to systematically evaluate the compositionality in **all of the three aspects**: sys, pro, sub, rather than sys only that has been investigated in previous works.
> > > > >
> > > > >   - provide **two comprehensive diagnosing benchmarks** that are **much closer to real-world applications** than the synthetic ColorMNIST that has been used in previous works.
> > > > >
> > > > > - Thirdly, our current empirical results also validate the above conclusion.
> > > > >
> > > > >   - The proposed evaluation metric $H_n$ on compositionality shows **high positive Pearson coefficients of 0.975 (on CGQA) and 0.971 (on COBJ) with our reference metric of $A_{non}$**, which can be regarded as your expected metric truly evaluating the CL ability of a feature extractor. Note that $A_{non}$ eliminates the effect of the classifier via evaluation on few-shot tasks whose labels are from $\mathcal{Y}_{tr}$ instead of $\mathcal{Y}_{sys}$.
> > > > >
> > > > >   - Beyond the high correlation with such a continual learning metric, the breakdown of $H_n$ into three proposed **multi-dimensional measures**, including $A_{sys}$, $A_{pro}$, and $A_{sub}$, offers a comprehensive evaluation of the compositionality and foothold for a detailed comparison between methods.
> > > > >
> > > > >   - Moreover, we highlight that both $H_n$ and $A_{non}$ are **indeed continual learning metrics**, which can be conducted at any checkpoint, though we report the performance after continual learning of all tasks. The result of $H_{n}(i)$ at the $i$-th task on the 10-way COBJ dataset is shown below, from which we conclude that methods tend to improve compositionality when seeing more labels (combinations of concepts).
> > > > >
> > > > >     | Methods   | $H_n(1)$ | $H_n(2)$ | $H_n(3)$ |
> > > > >     | --------- | -------- | -------- | -------- |
> > > > >     | Finetune  | 37.57    | 38.32    | 37.82    |
> > > > >     | ER        | 30.27    | 33.33    | 36.99    |
> > > > >     | GEM       | 38.05    | 37.60    | 37.36    |
> > > > >     | LwF       | 38.13    | 43.34    | 45.19    |
> > > > >     | EWC       | 37.34    | 36.86    | 38.68    |
> > > > >     | Finetune* | 37.41    | 37.07    | 40.93    |
> > > > >     | ER*       | 31.11    | 37.30    | 38.66    |
> > > > >     | GEM*      | 37.45    | 35.60    | 40.93    |
> > > > >     | LwF*      | 37.84    | 44.01    | 44.75    |
> > > > >     | EWC*      | 30.76    | 34.18    | 36.20    |
> > > > >
> > > > >
> > > > >
> > > > > **References**:
> > > > >
> > > > > 1. Li, Y., Zhao, L., Church, K., & Elhoseiny, M. (2020). Compositional language continual learning. ICLR.
> > > > >
> > > > > 2. Biesialska, M., Biesialska, K., & Costa-Jussa, M. R. (2020). Continual lifelong learning in natural language processing: A survey. COLING.
> > > > >
> > > > > 3. Zhang, Y., Wang, X., & Yang, D. (2022). Continual sequence generation with adaptive compositional modules. ACL.
> > > > >
> > > > > 4. Mendez, J. A., & Eaton, E. (2021). Lifelong learning of compositional structures. ICLR.
> > > > >
> > > > > 5. Ostapenko, O., Rodriguez, P., Caccia, M., & Charlin, L. (2021). Continual learning via local module composition. NeurIPS.
> > > > >
> > > > > 6. Veniat, T., Denoyer, L., & Ranzato, M. A. (2021). Efficient continual learning with modular networks and task-driven priors. ICLR.

---

> > ### Comment · Reviewer_fvLh · 2023-08-21
> >
> > Thanks for the authors' clarification.
> >
> > Q1: As suggested by authors, Acon is not a suitable metrics to evaluate the stability-plasticity dilemma. Then _what is the quantitative metrics that truly evaluate the continual learning (CL) ability of a feature extractor?_
> >
> > This evaluation metrics is critical to validate the paper's claim.
> > To verify the main paper's contribution "compositionality addresses the stability-plasticity dilemma", the paper has to include this CL evaluation metrics and analyze its correlation with the compositionality metrics, i.e., Hn.
> > Otherwise, we cannot make a meaningful conclusion that "compositionality addresses the stability-plasticity dilemma". Only inspecting few qualitative visualization is not sufficient to make such a conclusion. Meaningful quantitative evaluations are needed.
> >
> > Q2: The motivations are solid. I suggest including these discussions in the paper.

---

### Official Review · Reviewer_9Qe4 · 2023-07-20
**Well designed and well explained benchmark and evaluation**

**Rating:** 8
**Confidence:** 5

**Strengths:**

1. The proposed benchmark is well-designed and addresses an interesting problem combining compositional learning and continual learning. There is a notable lack of such benchmarks in the literature.
2. The experimental evaluation is thorough and well-explained. The chosen metrics cover a spectrum of aspects of interest: "standard" continual learning performance (on tasks seen during training), three "compositional" abilities, and two "reference" or non-compositional abilities.
3. The details of the benchmark creation process are covered sufficiently in Appendix C.

**Additional Feedback:**

The following points are provided as feedback to hopefully help better shape the submitted manuscript, but did not impact my recommendation in a major way.

Sec 3
- I find the notion of pasting images together in CGQA to create labels that are combinations of concepts a bit odd (it's super synthetic), but it could be interesting
- One downside is that the feature spaces representing each concept are completely disentangled, which isn't the case for real images

Sec 4
- How is the non-novel testing accuracy different from the training tasks accuracy if the label is from the training set of labels? Is it that the tasks themselves contain training labels but the combination of K labels is not in the set of training tasks?

Sec 5
- In this section, I would have preferred o read some description of the process used to generate the compositional data sets rather than the motivation (or maybe just a more compressed motivation and the description).

Sec 6
- The experimental evaluation is well explained and well designed
- It seems that there are errors in the result text?
    - Multitask isn't the best on H_n (RPSnet and ER* are better in COBJ).
    - MNTDP* is not the top performer on Hn on COBJ, only on Acon. Is this "outperforming" the others?

Typos
- Tons of grammar errors and typos

**Clarity:**

The paper is mostly clearly written, except for a number of typos and grammar errors.

**Correctness:**

The benchmark appears to be correct and the choice of baselines and metrics is appropriate.

**Documentation:**

The documentation of the benchmark appears to be sufficient.

**Ethics:**

No.

**Limitations:**

The discussion of limitations is quite short and superficial.

**Opportunities For Improvement:**

1. The related work section in the main paper is "inadequate" at best. The authors do provide a more complete literature survey in Appendix B, which I strongly encourage them to move to the main paper.
2. The strict requirement that the feature extractor should be frozen is insufficiently justified. In particular, it seems to me that the _benchmark_ shouldn't impose restrictions on the choices the _methods_ can employ, unless such choices would require additional assumptions (e.g., knowledge about "hidden" information about the benchmark). This doesn't seem to be the case for this work. What if some new method uses feature finetuning as a means to learn new compositional tasks? What about approaches that don't explicitly separate feature learning from classifier learning?
3. The proposed evaluation setting doesn't appear to be strictly continual, in the "never-ending" sense. The training is certainly sequential, but there is a distinct separation between the continual training phase and the fewshot evaluation phase. One downside of this is that it calls into question the need to train sequentially: if the learner will be allowed to see all training tasks before actually being evaluated for its compositional abilities, couldn't it just store all the data and wait until the end and perform multitask training? Would a more natural setting without this explicit separation between continual training and evaluation be possible? If not, what is a motivating example for a setting where continual training is required prior to few-shot testing deployment?

**Relation To Prior Work:**

This is one of the main weaknesses of the paper, and should be addressed in a revision by including the discussion from the Appendix in the main paper.

**Summary And Contributions:**

The authors propose two benchmarks for evaluating compositionality of continual learners (CGQA and COBJ) and an evaluation protocol (compositional few-shot testing, CFST) to assess the continual learners' ability to discover compositional concepts from a task sequence. A comprehensive evaluation of baselines on the proposed benchmarks sheds light on existing methods' compositional capabilities.

---

**Post-rebuttal update**

Per the authors responses and updates to their manuscript, my main concerns have been addressed and I have increased my rating from 7 (accept) to 8 (clear accept).

---

> ### Author Response · Authors · 2023-08-17
> **To Reviewer 9Qe4 (1/2)**
>
> We sincerely appreciate your constructive comments on this paper. We detail our response below point by point. Please kindly let us know if our response addresses the issues you raised in this paper.
>
> ### Q1: Move related work from appendix to main paper
> >
> > - This is a very good advice. Unfortunately, due to the page limit, we did not include them in our main text. However, we clearly understand that they are important for the reader so we will include them in the revision.
>
> ### Q2: Too strict requirement of frozen feature extractor
> >
> > First I would like to highlight that our benchmarks **do not impose any restriction** on the methods. Maybe our claim in the main paper misleads you, and we will correct the expression in the revision.
> >
> > - During **CL training**, the methods’ feature extractor does not have to be frozen (which is also the case in our experiments).
> >
> > - We now explain why we chose to use **few-shot tasks** and **frozen feature extractors** during **evaluation**.
> >
> >   - At the specific checkpoint (after finishing all continual training tasks), we use our evaluation tasks to evaluate the model’s compositionality. In such a condition, if the number of support samples in a evaluation task is large, the feature extractor may learn from them, and thus, we can not actually judge whether the good performance comes from the original feature extractor (learned from old tasks).
> >
> >   - We also have tried to not freeze the feature extractor and the results are presented in Appendix E.7. For your convenience, we show our observation: all methods showed a performance drop if not freezing the feature extractor, especially for ER. It was clearly an overfitting issue and this bad effect was method-dependent. So in order to eliminate this effect when comparing the accuracies among methods, we froze the feature extractor.
> >
> > - Note that ``frozen feature extractor'' is not a strict requirement, thus, methods that don't explicitly separate feature learning from classifier learning can also use our evaluation protocol.
>
> ### Q3: ``Never-ending'' CL sense
> >
> > This is a very good comment. Our reported results were indeed static (only evaluating after finishing all continual training tasks), but the setting is clearly a never-ending CL.
> >
> > - We claim that our diagnosing evaluation can be used at any checkpoint (not to be restricted at the end of all continual training tasks). We report the per-task Hn on 10-way COBJ as follows:
> >
> >    | Methods | Hn_1 | Hn_2 | Hn_3 |
>     | -----------  | ---      | ---      | ---     |
>     | Finetune  | 37.57 | 38.32 | 37.82 |
>     | ER        | 30.27 | 33.33 | 36.99 |
>     | GEM       | 38.05 | 37.60 | 37.36 |
>     | LwF       | 38.13 | 43.34 | 45.19 |
>     | EWC       | 37.34 | 36.86 | 38.68 |
>     | Finetune* | 37.41 | 37.07 | 40.93 |
>     | ER*       | 31.11 | 37.30 | 38.66 |
>     | GEM*      | 37.45 | 35.60 | 40.93 |
>     | LwF*      | 37.84 | 44.01 | 44.75 |
>     | EWC*      | 30.76 | 34.18 | 36.20 |
> >
> >    - where Hn_1 denotes Hn after finishing the first continual task. We deepcopy the feature extractor and train a new classifier for each compositional testing task.
> >
> >    - A rough observation: methods tend to improve compositionality when seeing more labels (combinations of concepts).
>
> ### Q4: Disentangled concepts are not real
> >
> > Thank you very much to point out the limitation. We apologize for not discussing more about this due to the page limit. We now explain why we choose those ''visual'' concepts to construct our benchmarks.
> >
> > - Note that, we want to achieve **balance** and **flexible** combinations of concepts. That is, the number of instances for different combinations of concepts can be similar (no long-tailed combinations) and we can combine any pair of concepts together.
> >
> > - However, some concepts like motion and human mood can only be combined with the human concept (maybe it is better to explain as ''attribute'' in this work). This limits the flexibility and limits the number of re-combinations in the systematicity test since it is difficult to find various kinds of moods on other animals (maybe cats can :p ).

---

> > ### Author Response · Authors · 2023-08-17
> > **To Reviewer 9Qe4 (2/2)**
> >
> > ### Q5: Relation to prior work
> > >
> > > Thank you very much to point out our weaknesses and we will address them in our revision.
> > >
> > > - We have mentioned in Sec 2 (Related Works) that some vision benchmarks evaluate compositionality (including works in the CL field) and we have pointed out that some benchmarks are toy and evaluate only systematicity.
> > >
> > > - As for the CL methods, unfortunately, due to the page limit, we apologize for not including a detailed discussion in our main paper. However, we clearly understand that they are important and will include them in the revision.
> > >
> > > - Additionally, as for the relationship with CZSL, we have illustrated the difference in Figure 1 and Remark 3.2.
> > >
> > > - As for the relationship between our evaluation protocol with forgetting, we have investigated empirically by a case study in Sec 6 [lines 276-304] and also have provided detailed experiments in Appendix E.4.
> >
> > ### Q6: Question about non-novel testing accuracy vs training accuracy in Sec 4
> > >
> > > - The answer is yes. Although one non-novel testing task contains the same number of labels as one training task, the K labels are **randomly chosen from the training label pool** and there is no intersection between their sample instances. That is, it is a small probability that the K labels of a non-novel testing task are just the same as that in one training task (of course, the number of training samples for each label is relatively smaller than that in the training tasks).
> >
> > ### Q7: Put more description of the construction process rather than the motivation in Sec 5
> > >
> > > - We apologize for not including a detailed construction process in our main paper, due to the page limit. However, we clearly understand that this is very important for the readers to understand our work, thus, we will compress the motivation and put the construction process to the main paper in the revision.
> >
> > ### Q8: Question about experimental results
> > >
> > > 1. Multitask is not the best on Hn in COBJ (RPSnet and ER* are the best)
> > >
> > >   - Firstly, we would like to highlight that Hn evaluates the **compositionality of the feature extractor**.
> > >
> > >   - This observation is quite interesting, that, Multitask may not necessarily be the upper bound in terms of compositionality. In CGQA, compositionality is easier to learn since we visually split the concepts which the models are expected to learn. Multitask shows a great superiority on Hn in CGQA, which is also consistent with the CAM visualization results in Appendix E.3 and Fig7. Multitask could recognize more concepts than Finetune.
> > >
> > >   - However, in the real-world case (COBJ), concepts are not as visually separable as that in CGQA. Our CAM visualization results in Appendix Fig8 showed that Multitask was better than Finetune but the gap was not so large (the number of recognized concepts by Multitask was larger but similar to that by Finetune). That is, Multitask might not necessarily beat CL methods in terms of compositionality.
> > >
> > >   - Hope my analysis solves your question.
> > >
> > > 2. MNTDP* is not the top performer on Hn on COBJ, only on Acon.
> > >
> > >   - As we highlighted in the above question, a model with better Acon does not necessarily have better Hn (compositionality).
> > >
> > >   - This result indicated that MNTDP* showed no superiority in compositionality. The high average test accuracy was due to the zero forgetting of old tasks since it froze all learned modules for old tasks.
> > >
> > > - The above two observations also indicate the shortcomings of average test accuracy Acon. **Our evaluation metric Hn provides more insights**.
> >
> > ### Q9: Grammar errors and typos
> > >
> > > - Sorry for my grammar errors and typos that lead to the misunderstanding of some parts.
> > >
> > > - We will carefully check those typos in our revision.
> >
> > Thank you again for your comments.

---

> > > ### Comment · Reviewer_9Qe4 · 2023-08-21
> > >
> > > I thank the authors for their responses, which adequately address my main concerns. I will raise my score and am increasingly convinced that this paper would be a valuable addition to the Datasets and Benchmarks track.

---

### Official Review · Reviewer_RpQ2 · 2023-07-21
**This is a very interesting benchmark for studying compositionality in continual learning, which is often overlooked.**

**Rating:** 8
**Confidence:** 4
**Correctness:** Yes.
**Clarity:** Yes, the paper is well written.

**Strengths:**

1. The paper is well-written and easy to follow. The proposed benchmark is reasonable and what it tests, to my knowledge, cannot be found in the existing literature.

2. The way the authors test what the agent learns during the CL phase is interesting, i.e., by testing its downstream performance without finetuning.

**Additional Feedback:**

I think this work proposes a novel but unique angle for research in continual learning, hence I recommend acceptance of it.

**Documentation:**

It is well documented, but the code is not yet provided.

**Ethics:**

No.

**Limitations:**

The main limitation is that all concepts are pertaining to visual entities, but in reality, there are concepts about things like motion, human mood, or anything that is not directly describable. I believe those will offer a quite different challenge for generalization.

**Opportunities For Improvement:**

It is not clear if model sizes are kept constant, which is especially important when you compare methods like RPSnet to other CL methods.

**Relation To Prior Work:**

Yes, it is clearly discussed.

**Summary And Contributions:**

This work proposes novel vision-based benchmarks (the synthetic CGQA and realistic COBJ) for studying how the agent learns compositional knowledge continually. The two benchmarks are designed to evaluate three compositional capabilities including systematicity, productivity, and productivity. The authors also evaluate both convnet and ViT-based architectures. In general, I think it is a good paper that presents a new challenge to the continual learning field, and the proposed benchmark serves a good fit for testing algorithms resolving the challenge.

---

> ### Author Response · Authors · 2023-08-17
> **To Reviewer RpQ2**
>
> We sincerely appreciate your constructive comments on this paper. We detail our response below point by point. Please kindly let us know if our response addresses the issues you raised in this paper.
>
> ### Q1: Model size comparison for RPSnet to other CL methods.
> >
> > This is a very good concern. We would like to highlight that we follow the **original implementation of modularity-based methods** which freeze old modules to maximally prevent catastrophic forgetting and **learn new modules sub-linearly** to solve new knowledge.
> >
> > - RPSnet (also MNTDP) exactly has much **more model parameters** than other methods when the current task id $t>0$.
> >
> > - However, in order to **guarantee fair comparison**, RPSnet only allows training one module on each layer for one task. Overall, **the number of trainable parameters remains the same and comparable to other baselines**.
>
> ### Q2: Limitation on selected concepts.
> >
> > This is a very good comment and advice. The reason we chose these ``visual'' concepts is as follows:
> >
> > - We wanted to achieve **balance** and **flexible** combinations of concepts. That is, the number of instances for different combinations of concepts should be similar (no long-tailed combinations) and we can combine any pair of concepts. However, some concepts, like motion and human mood, can only be combined with the human concept (maybe it is better to explain as ``attribute'' in this work). This limits the flexibility and limits the number of re-combinations in the systematicity test since it is difficult to find various kinds of moods on other animals (maybe cats can :p ).
> >
> > - Our future work is to further study concepts hierarchically to introduce different generalization challenges. We can contain more ``abstract'' (not visual) concepts, e.g., by specifically combining facial or gesture benchmarks.
>
> Thank you again for your comments.

---

### Official Review · Reviewer_xdij · 2023-07-22
**Comments to Submission 436**

**Rating:** 4
**Confidence:** 4
**Clarity:** Yes. The paper is written in a clear …

**Strengths:**

The CGQA and COBJ benchmarks are designed to test the learner's ability to handle novel combinations of seen concepts, providing a comprehensive evaluation of the compositional generalization capability of continual learning methods.

The paper evaluates three compositional capabilities: systematicity, productivity, and substitutivity.

The benchmarks do not contain any sensitive data, such as medical data, which makes them ethically sound for use in research.

The baseline methods are all public and widely used, which means they are accessible to researchers and can be used to compare the performance of different methods.


**Additional Feedback:**

The authors are recommended to address my concerns listed in the 'Opportunities For Improvement' and 'Limitation' parts.

**Correctness:**

The reviewer has concerns about whether the modules for old tasks shall be frozen.

**Documentation:**

Yes.

**Ethics:**

No.

**Limitations:**

The authors discussed the limitation of the work from the technology point of view in Section 7. Nonetheless, I don't find an explicit discussion about the negative societal impact. The risk of misusing the dataset may be discussed.

**Opportunities For Improvement:**

I was very concerned about the construction process of CFST. CFST is like a simple multi-label recognition task, where the image is divided into several parts, each part has an object or concept, and the network is required to predict all labels. I didn't see how this setting could reflect the difference between the task of compositional generalization and traditional multi-label classification tasks. It's clear that this combination dataset does not quite meet the requirements of real-world scenarios and the task of compositional generalization. The author can provide a better explanation and point out the difficulties of this task.

CGQA may deviate too much from practical applications. CGQA actually divides the picture into a grid and places a picture in each grid, then treats the predicted label as the model's compositional ability. This approach does not reflect the requirements of this task. In fact, this task trains the network to classify each grid and uses the final combination as the multi-label prediction result. COBJ seems to be more in line with reality, but I need more examples to demonstrate this benchmark. At present, this dataset seems like the object detection dataset, but it does not require location cues. Therefore, I cannot judge whether this task has significant practical significance.

The process of freezing the modules for old tasks may not be appropriate for new tasks unless the modules are compositional enough. This could limit the adaptability of the methods to new tasks.

The paper mentions that large efforts are still needed on modularity-based methods for not bringing benefits to compositionality. This suggests that these methods may not be fully developed or optimized yet. But as the requirements of dataset track, this is not a big problem.



**Relation To Prior Work:**

The difference from previous datasets was described in Sections 1 and 2.

**Summary And Contributions:**

The paper proposes an interesting problem of continual compositional generalization. It introduces two new benchmarks, CGQA and COBJ, to evaluate the performance of continual learning algorithms in extracting compositional features from a sequence of tasks.

CGQA (Compositional Generalization Question Answering) is a synthetic vision benchmark designed to evaluate the compositional generalization capability of continual learning methods. It is built upon the GQA dataset and is designed to test the learner's ability to handle novel combinations of seen concepts. COBJ is a real-world continual vision benchmark. The source dataset is Objects365. Objects365 is a high-resolution dataset designed for object detection with enough combinations of objects in the wild.

The paper also introduces five few-shot testing schemes. These schemes are designed to provide a comprehensive view of the learners' compositionality. The paper compares several baseline methods for continual learning, including GEM (Gradient Episodic Memory), LwF (Learning without Forgetting), EWC (Elastic Weight Consolidation), ER (Experience Replay), MNTDP (Modular Networks with Task-Dependent Priors) and RPSnet. In the class-IL setting, where the task identifier is unknown, a single-head classifier is used for these methods. In the task-IL setting, where the task identifier is known, a multi-head classifier is used instead. All these methods are evaluated with a ResNet-18 backbone.

The paper conducts comprehensive computational experiments to answer several key questions about the performance of continual learning algorithms. These questions include whether the proposed benchmarks can correctly evaluate learners' performance in extracting compositional features, whether the five few-shot testing schemes can provide an informative view of learners' compositionality, and whether modularity-based approaches have better few-shot testing accuracy, indicating that they are good at extracting compositional features.

---

> ### Author Response · Authors · 2023-08-17
> **To Reviewer xdij**
>
> We sincerely appreciate your constructive comments on this paper. We detail our response below point by point. Please kindly let us know if our response addresses the issues you raised in this paper.
>
> ### Q1: Differences between CFST and multi-label classification tasks
> >
> > We would like to highlight the key difference between our CFST and the multi-label recognition task, which we have already illustrated in Figure 1 and Remark 3.2 [Lines 106-111] in the main text.
> >
> > - In CFST **only the image-level label is available and no concept label is provided**, which is much more challenging than multi-label classification where labels of all concepts in an image are accessible.
> >
> > - Thus, CFST requires **inference of the underlying compositional concepts behind an image**, while multi-label classification does not encourage compositional learning as we have detailed in Appendix A.
> >
> >   - Capturing those discriminative features only for classification (e.g., the shape that differentiates "horse" from "human") and failing to capture all the compositional features (e.g., the fine-grained features helpful for identification of "horse") likely struggles in a future CFST task (e.g., classifying between "zebra" and "horse").
>
> ### Q2: How CGQA and COBJ reflect practical applications
> >
> > The objective of our benchmark is exactly to evaluate the capability of a model in compositional learning, i.e., **whether it understands the previously learned concepts and generalizes to future tasks**. We underline that CGQA and COBJ meet this objective in practice:
> >
> > - First, we would like to humbly clarify that the task **does not** "train the network to classify each grid", as concept labels are not accessible (see the response to Q1).
> >
> > - Thus, a combined image with only a single label in CGQA is **the same as a natural image in practical applications**, except that we aim to construct a dataset like CGQA with **(1)** all the concepts in an image **easy to parse and interpret**, and **(2)** **a comparably small number of concepts**. CGQA constructed in such a manner serves as a **less challenging benchmark** to evaluate various methods.
> >
> > - The **more challenging benchmark of COBJ** contains images that also have **single labels without concept/object labels**; compared to CGQA, the concepts within each image are more complicated.
> >
> >   - We have detailed the construction process of COBJ in Appendix C;
> >
> >   - We have provided image examples in Appendix Figure 5.
> ​
> ### Q3: Freezing modules limits the adaptability of the methods to new tasks
> >
> > - We follow the **original implementation of modularity-based methods** which **freeze old modules to maximally prevent catastrophic forgetting** and learn new modules sub-linearly to solve new knowledge.
> >
> > - Freezing modules is also crucial to **guarantee fair comparison**, so that for each task the number of trainable parameters remains the same and comparable to other baselines. In fact, we had previously jointly trained all modules of RPSnet, resulting in the performance of $H_n=67.99$ on CGQA. Though this is higher than that by freezing old modules ($H_n=59.94$ in our paper), the improvement by greater model capacity is not fair.
>
> ### Q4: No discussion about the negative societal impact
> >
> > - Thank you very much to point out the limitations. Our benchmarks do not contain any sensitive data, such as medical data, which makes them ethically sound for use in research.
> > - We will update the discussion in our new revision as soon as possible.
>
> Thank you again for your comments.

---

> ### Author Response · Authors · 2023-08-26
> **Please kindly let us know whether you have further concerns.**
>
> Hi Reviewer xdij,
>
> We would like to follow up to see if our response addresses your concerns or if you have any further questions. We would really appreciate the opportunity to discuss this further if our response has not already addressed your concerns. Thank you again!

---

### Author Response · Authors · 2023-08-26
**Response to all reviews**

Thank you for the reviews. We summarized some updates based on the suggestions and the changes in revision were highlighted in blue.
>
> - Abstract: Explained the acronyms (i.e., CGQA, COBJ, CFST).
> - Fig 1: Clarified hidden and given labels and added the ''one-hot label'' mark.
> - Rearranged Sec 2 Related Works: Moved continual learning methods from the Appendix to the main text and included more papers suggested by the reviewers.
> - Sec 3: Improved Remark3.2 and clarified the label space and notations. **Specifically, We separated the definitions of labels $y$ and concept combinations $c^{1:M}$.**
> - Sec 4: Revised the writing and removed some redundant texts.
> - Sec 5: Moved the construction process of CGQA and COBJ from the Appendix to the main text.
> - Sec 7: added discussion about the societal impact.

---

### Decision · Program_Chairs · 2023-09-22

**Decision:**

Accept (Poster)

**Comment:**

This paper underwent a thorough review process by five reviewers, with most of them showing a positive inclination toward accepting the paper after the authors' rebuttal. Based on the positive feedback from the majority of reviewers, I am inclined to recommend accepting this paper. I encourage the authors to carefully incorporate the suggested revisions and clarifications from the reviewers (especially for the concern of how CGQA and COBJ reflect practical applications raised by the reviewer xdij) to further enhance the paper's quality in the final version.